# TRIM16 controls assembly and degradation of protein aggregates by modulating the p62-NRF2 axis and autophagy

Kautilya Kumar Jena[1,2], Srinivasa Prasad Kolapalli[1,†], Subhash Mehto[1,†], Parej Nath[1,2], Biswajit Das[3,4], Pradyumna Kumar Sahoo[1], Abdul Ahad[5], Gulam Hussain Syed[6], Sunil K Raghav[5], Shantibhusan Senapati[3], Swati Chauhan[7] & Santosh Chauhan[1,*]

## Abstract

Sequestration of protein aggregates in inclusion bodies and their subsequent degradation prevents proteostasis imbalance, cytotoxicity, and proteinopathies. The underlying molecular mechanisms controlling the turnover of protein aggregates are mostly uncharacterized. Herein, we show that a TRIM family protein, TRIM16, governs the process of stress-induced biogenesis and degradation of protein aggregates. TRIM16 facilitates protein aggregate formation by positively regulating the p62-NRF2 axis. We show that TRIM16 is an integral part of the p62-KEAP1-NRF2 complex and utilizes multiple mechanisms for stabilizing NRF2. Under oxidative and proteotoxic stress conditions, TRIM16 activates ubiquitin pathway genes and p62 via NRF2, leading to ubiquitination of misfolded proteins and formation of protein aggregates. We further show that TRIM16 acts as a scaffold protein and, by interacting with p62, ULK1, ATG16L1, and LC3B, facilitates autophagic degradation of protein aggregates. Thus, TRIM16 streamlines the process of stress-induced aggregate clearance and protects cells against oxidative/proteotoxic stress-induced toxicity *in vitro* and *in vivo*. Taken together, this work identifies a new mechanism of protein aggregate turnover, which could be relevant in protein aggregation-associated diseases such as neurodegeneration.

**Keywords** autophagy; NRF2; p62; protein aggregates; TRIM16
**Subject Categories** Autophagy & Cell Death; Post-translational Modifications, Proteolysis & Proteomics
**The EMBO Journal (2018) 37: e98358**

## Introduction

Misfolded proteins are a common outcome of protein biosynthesis, and about 30% of newly synthesized proteins end up misfolded (Schubert *et al*, 2000). Normally, these misfolded proteins are ubiquitinated and degraded by the proteasome system and/or autophagy process (Kraft *et al*, 2010; Lamark & Johansen, 2012; Amm *et al*, 2014; Jackson & Hewitt, 2016). Multiple factors ranging from genetic mutations to cellular (oxidative/proteotoxic stress) and environmental stresses can trigger protein misfolding overwhelming the cellular capacity to clear them leading to their aggregation and accumulation (Goldberg, 2003; Tyedmers *et al*, 2010). Several lines of evidence suggest that formation of protein aggregates is a cytoprotective phenomenon and perturbation in this process results in accumulation of misfolded proteins implicated in many pathological conditions collectively termed as proteinopathies (Alzheimer's, Parkinson's, amyotrophic lateral sclerosis, etc.; Arrasate *et al*, 2004; Tanaka *et al*, 2004; Douglas *et al*, 2008; Tyedmers *et al*, 2010). As the unrestrained increase in misfolded proteins is deleterious, the unresolved increase in protein aggregates is also cytotoxic, and clearance of these aggregates increases the cell life span (Komatsu *et al*, 2006; Sarkar *et al*, 2007; Kirkin *et al*, 2009).

Two types of protein aggregates are documented. The inhibition of the ubiquitin–proteasome system leads to the formation of protein aggregates named as aggresomes (Johnston *et al*, 1998; Olzmann *et al*, 2008), and exposure to other physiological stresses, such as oxidative stress, protein translation inhibition, and heat shock, results in the formation of another kind of protein aggregates termed as ALIS (aggresome-like induced structures; Szeto *et al*, 2006). The formation of aggresomes but not of ALIS is dependent on microtubules and dynein motors (Johnston *et al*, 2002; Szeto *et al*, 2006). Both types of protein aggregates are considered to be

1 Cell Biology and Infectious Diseases Unit, Institute of Life Sciences, Bhubaneswar, India
2 School of Biotechnology, KIIT University, Bhubaneswar, India
3 Tumor Microenvironment and Animal Models, Institute of Life Sciences, Bhubaneswar, India
4 Manipal University, Manipal, India
5 Immuno-Genomics and Systems Biology, Institute of Life Sciences, Bhubaneswar, India
6 Molecular Virology and Infectious Diseases, Institute of Life Sciences, Bhubaneswar, India
7 Translational Research, Institute of Life Sciences, Bhubaneswar, India
*Corresponding author. Tel: +91 674 2304334; E-mail: schauhan@ils.res.in
†These authors contributed equally to this work

degraded by selective autophagy, termed as aggrephagy (Kirkin *et al*, 2009). The defects in autophagy lead to the accumulation of protein aggregates and have been linked to severity of several proteinopathies and neuropathies (Hara *et al*, 2006; Komatsu *et al*, 2006, 2007b; Yao, 2010).

NRF2 (nuclear factor erythroid 2-related factor 2) is a key transcription factor that drives the cellular response to combat a variety of stresses including oxidative stress, proteotoxic stress and electrophilic insults (Takaya *et al*, 2012; Jaramillo & Zhang, 2013; Kansanen *et al*, 2013). Under basal conditions, KEAP1 (Kelch-like ECH-associated protein 1) interacts with NRF2 and mediates its degradation (Kansanen *et al*, 2013). Exposure of cell to oxidative stress and electrophilic insult inactivate KEAP1 leading to the nuclear translocation of stable NRF2 followed by transcriptional induction of cytoprotective and antioxidant response genes (Motohashi & Yamamoto, 2004). Multiple studies have shown the role of autophagy adaptor protein p62/SQSTM1 (sequestosome 1, hereafter referred to as p62) in positive regulation of NRF2 by either interfering with NRF2-KEAP1 interaction or facilitating the sequestration and autophagic degradation of KEAP1 (Copple *et al*, 2010; Jain *et al*, 2010; Komatsu *et al*, 2010; Riley *et al*, 2011). Subsequently, the free NRF2 translocates to the nucleus and upregulates expression of antioxidant response element (ARE) containing genes such as *Hmox1*, *Nqo1*, and *p62* (Copple *et al*, 2010; Jain *et al*, 2010; Komatsu *et al*, 2010; Lau *et al*, 2010). As a positive feedback mechanism, p62 stabilizes NRF2 leading to further amplification of the NRF2 response (Jain *et al*, 2010). The p62 is implicated in both assembly and degradation of protein aggregates (Komatsu *et al*, 2007b; Pankiv *et al*, 2007; Nezis *et al*, 2008). It interacts with ubiquitin over the misfolded proteins, and due to its capacity to self-polymerize, it facilitates the formation of protein aggregates (Nagaoka *et al*, 2004; Seibenhener *et al*, 2004; Bjorkoy *et al*, 2005, 2006). Subsequently, p62 interacts with LC3B and targets protein aggregates to autophagic degradation (Bjorkoy *et al*, 2006).

The ubiquitin system plays a crucial role in formation and degradation of protein aggregates (Dodson *et al*, 2015). The chronological actions of three enzymes catalyze the process of protein ubiquitination: ubiquitin-activating enzymes, ubiquitin-conjugating enzymes, and ubiquitin ligases, known as E1s, E2s, and E3s. The proteotoxic and oxidative stresses disrupt protein homeostasis leading to increase ubiquitin-tagged misfolded proteins which subsequently form protein aggregates.

Tripartite motif (TRIM) proteins are a family of about 75 proteins with E3 ligase activities and with varied functions in cellular processes, including innate immunity, cell proliferation, differentiation, development, carcinogenesis, apoptosis, and autophagy (Nisole *et al*, 2005; Hatakeyama, 2017). Several of the TRIM proteins are shown to function as autophagy regulator–receptor and modulate the selective autophagy process (Mandell *et al*, 2014a,b; Kimura *et al*, 2015, 2016a,b; Chauhan *et al*, 2016). TRIMs are shown to form a platform for the assembly of core autophagic machinery, including autophagy initiation (ULK1), elongation (ATG16L1), and completion (LC3B) factors and facilitate the interaction of the autophagic machinery with the specific autophagic cargo. In a previous study, we have shown that TRIM16 interacts with galectin-3 and recognizes damaged endomembranes leading to the assembly of autophagy machinery for selective sequestration of damaged lysosomes and phagosomes (Chauhan *et al*, 2016). More

recently, TRIM16 is shown to play a role in secretory autophagy where it interacts with SNARE proteins to recruit cargoes to autophagosomes (Kimura *et al*, 2016b).

In this study, we show that under oxidative or proteotoxic stress conditions, an intricate positive feedback loop involving TRIM16, NRF2, KEAP1, and p62 direct the formation of protein aggregates from misfolded proteins. TRIM16 regulates expression and stability of NRF2, KEAP1, and p62. For the first time, this study underscores the role of NRF2 in protein aggregate formation. Furthermore, we show that TRIM16 also drives the autophagic degradation of the protein aggregates. TRIM16 performs a comprehensive role in protein aggregates turnover to safeguard the cells from stress-induced highly toxic misfolded proteins both *in vitro* and *in vivo*.

# Results

## TRIM16 is required for biogenesis of aggresomes/ALIS

Previous studies indicate that TRIM16 is an oxidative stress-induced gene (Noguchi, 2008; Python *et al*, 2009; Hirose *et al*, 2011). Given the role of TRIM proteins including TRIM16 in selective autophagy (Mandell *et al*, 2014a,b, 2016; Kimura *et al*, 2015, 2016a,b, 2017a,b; Chauhan *et al*, 2016; Fraiberg & Elazar, 2016; Kumar *et al*, 2017), we envisaged the role of TRIM16 in autophagic degradation of protein aggregates (aggrephagy) formed during oxidative or proteotoxic stress conditions. The protein aggregates can be scored by immunofluorescence using anti-ubiquitin antibody (Kirkin *et al*, 2009; Lamark & Johansen, 2012; Svenning & Johansen, 2013; Lystad & Simonsen, 2015). If TRIM16 mediates the degradation of ubiquitinated protein aggregates, ubiquitin (hereafter, Ub) dots should accumulate in TRIM16-depleted cells. Surprisingly, as compared to control cells, the TRIM16 CRISPR-CAS9 knockout (hereafter TRIM16$^{KO}$; Appendix Fig S1A) or siRNA knockdown cells (Appendix Fig S1B and C) showed substantially reduced levels of Ub dots formed under oxidative ($H_2O_2$) or proteotoxic stress conditions (MG132, puromycin) (Fig EV1A–C, Appendix Fig S1D and E). In the case of puromycin treatment, the difference was observed in size (smaller) and numbers of Ub dots per cells (Fig EV1A and C) but not in the number of cells with Ub dots (Fig EV1B). The TRIM16$^{KO}$ HeLa cells were described previously (Chauhan *et al*, 2016; Appendix Fig S1A).

The p62 protein interacts with ubiquitinated misfolded proteins and promotes aggresomes/ALIS formation (Seibenhener *et al*, 2004; Bjorkoy *et al*, 2006; Johansen & Lamark, 2011). We next analyzed the status of Ub-p62 double-positive ALIS/aggresomes formed during oxidative ($H_2O_2$ and $As_2O_3$) or proteotoxic stress (MG132 and puromycin) conditions. As compared to control cells, TRIM16-depleted cells (TRIM16$^{KO}$ or TRIM16-siRNA) formed substantially reduced Ub-p62 double-positive aggresomes/ALIS (Figs 1A–F and EV1D–F). Aggresomes are present in the detergent-insoluble fraction of the cell lysate (Lystad & Simonsen, 2015). Next, we analyzed the status of aggresomes by Western blotting the detergent-soluble and detergent-insoluble cell fractions. The poly-ubiquitination of soluble proteins and soluble p62 was slightly lower in TRIM16$^{KO}$ cells compared to control cells (Fig 1G–I). However, there was a substantial reduction in insoluble (aggresome-enriched) poly-Ub or p62 in TRIM16$^{KO}$ cells compared to control cells (Fig 1G–I).

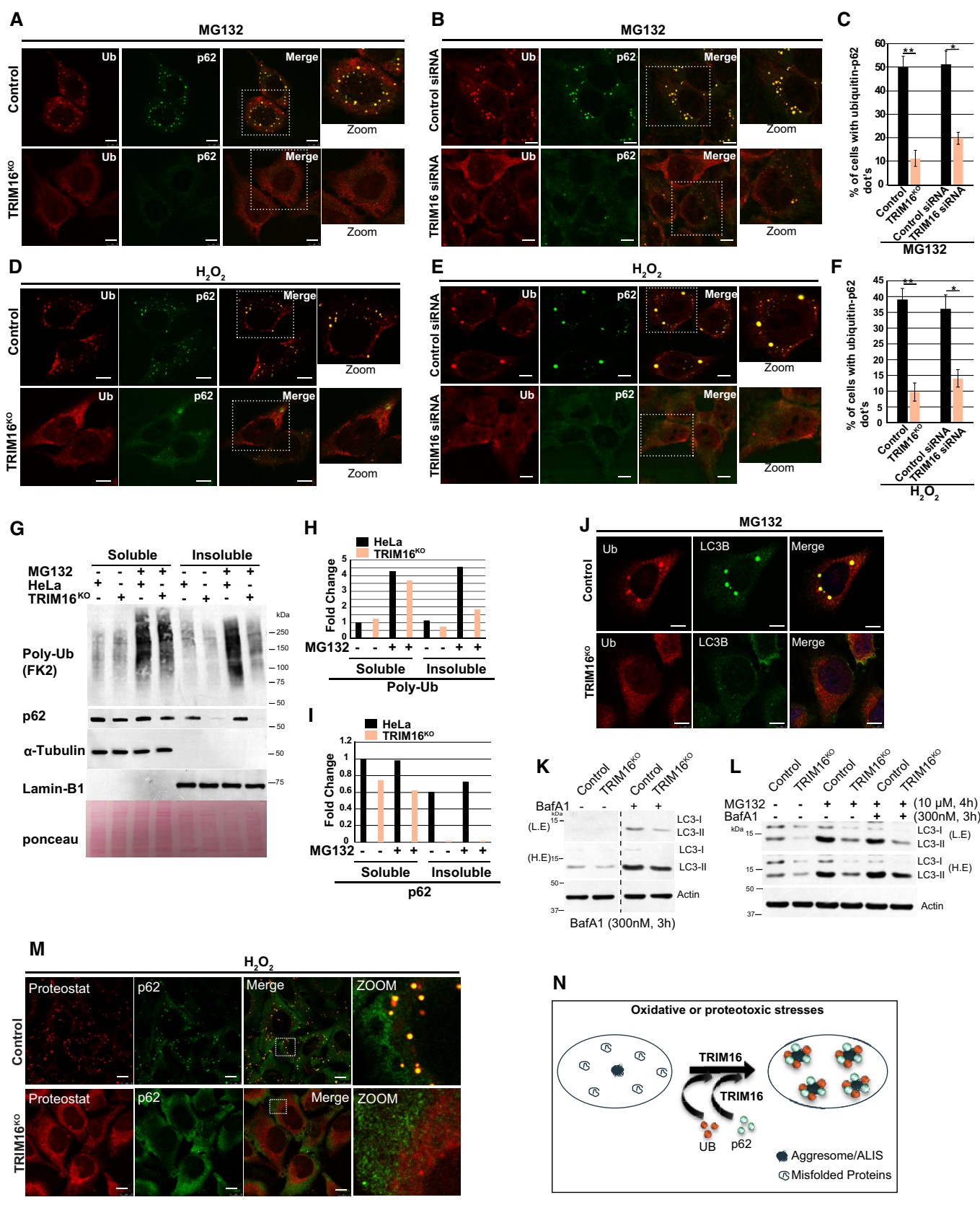

**Figure 1.**

**Figure 1.   TRIM16 is required for assembly of protein aggregates.**

A–F  (A, D) Representative confocal images of HeLa and TRIM16$^{KO}$ cells treated with (A) MG132 (20 μM, 2 h, scale bar: 7.5 μm) or (D) H$_2$O$_2$ (200 μM, 2 h) and the samples were processed for IF analysis with Ub and p62 antibody. (B, E) Representative confocal images of control siRNA and TRIM16 siRNA transfected cells treated with (B) MG132 (20 μM, 2 h), (E) H$_2$O$_2$, (200 μM, 2 h), where IF analysis was conducted with Ub and p62 antibodies. (C, F) The graph shows the percentage of cells with Ub-p62 co-localized dots. Data from ≥ 10 fields (40×), $n$ = 3, mean ± SD, **$P$ < 0.0003, *$P$ < 0.002 (Student's unpaired $t$-test).

G  Western blot (WB) analysis of detergent-soluble and detergent-insoluble fractions of HeLa and TRIM16$^{KO}$ cells treated with MG132 (50 μM, 1 h) and probed with indicated antibodies.

H, I  Densitometric analysis (using ImageJ) of WB normalized with α-tubulin or lamin-B1.

J  Representative confocal images of HeLa and TRIM16$^{KO}$ cells treated with MG132 (20 μM, 2 h) and processed for IF analysis with Ub and LC3B antibody.

K, L  WB analysis of LC3B levels in lysates of HeLa and TRIM16$^{KO}$ cells treated with (K) bafilomycin A1 (300 nM, 3 h) alone or (L) with MG132 (10 μM, 4 h) as indicated. L.E, low exposure; H.E, high exposure.

M  Representative confocal images HeLa and TRIM16$^{KO}$ cells treated with H$_2$O$_2$ (200 μM, 2 h) and processed for IF analysis with ProteoStat dye, and p62 antibody. To show the cellular details, the brightness of TRIM16$^{KO}$ cells images was increased more than control cells.

N  Pictorial representation of data presented in Figs 1 and EV1.

Data information: Unless otherwise stated, scale bar: 10 μm.
Source data are available online for this figure.

LC3B is a marker protein for autophagosomes (Klionsky *et al*, 2016). p62 binds to ubiquitinated misfolded proteins and recruits LC3B leading to the autophagy-mediated degradation of ubiquitinated proteins/protein aggregates (Vadlamudi *et al*, 1996; Pankiv *et al*, 2007). Since Ub-p62 double-positive dots were substantially reduced in TRIM16-depleted cells, we next analyzed the status of LC3B. We found that the oxidative and proteotoxic stress-induced Ub-LC3B double-positive dots were also significantly reduced in TRIM16-depleted cells as compared to the control cells (Figs 1J and EV1G and H).

Two alternative hypotheses were tested to understand the reason for fewer Ub-p62 or Ub-LC3B dots in TRIM16-depleted cells. The first hypothesis was that TRIM16 is a negative regulator of autophagy, and therefore, enhanced autophagy in TRIM16-depleted cells leads to rapid degradation of the aggregated proteins resulting in lesser Ub/p62/LC3B dots in the cells. To test this hypothesis, the status of autophagy was scrutinized in TRIM16-depleted cells using LC3B marker protein. We observed that basal and MG132-induced autophagy flux was attenuated in TRIM16$^{KO}$ cells compared to the control cells (Fig 1K–L). Conversely, overexpression of TRIM16 increased the LC3B levels in HEK293T cells (Fig EV1I and J). These data show that TRIM16 is a positive regulator of autophagy process, and thus, reduction in Ub/p62/LC3B dots in TRIM16-depleted cells is not due to enhanced autophagic flux.

The second hypothesis we tested was that a lesser number of aggresomes/ALIS are formed in TRIM16-depleted cells, leading to reduced number of Ub-p62 and Ub-LC3B dots. To evaluate this hypothesis, we employed an extensively used protein aggregates specific molecular rotor dye named as ProteoStat (Shen *et al*, 2011; Bershtein *et al*, 2013; Navarro & Ventura, 2014; Seo *et al*, 2016), which binds to the tertiary structure of aggregated proteins and emits red fluorescence. H$_2$O$_2$-treated HeLa cells showed red fluorescent protein aggregates which completely co-localize with p62, whereas TRIM16$^{KO}$ cells were almost devoid of protein aggregates (Figs 1M and EV1K, Appendix Fig S1F) showing that TRIM16 is important for biogenesis of protein aggregates. Taken together, the evidence suggests that TRIM16 is required for the assembly of protein aggregates (Fig 1N).

## TRIM16 regulates the p62-KEAP1-NRF2 system

Next, we investigated the molecular mechanism/s by which TRIM16 mediates biogenesis of protein aggregates. The p62 is known to play a vital role in the biogenesis of protein aggregates (Komatsu *et al*, 2007b). It regulates NRF2 which play a critical role in proteostasis and redox homeostasis (Komatsu *et al*, 2010; Lau *et al*, 2010; Kansanen *et al*, 2013; Wang *et al*, 2013; Pajares *et al*, 2017). We observed diminished levels of p62 in TRIM16-depleted cells (Fig 2A, Appendix Fig S2A) and the overexpression of TRIM16 resulted in

**Figure 2.   TRIM16 regulates the p62-KEAP1-NRF2 system.**

A  Western blot analysis of HeLa and TRIM16$^{KO}$ cell lysates probed with indicated antibodies.

B  Western blot analysis of lysates from HeLa and TRIM16$^{KO}$ cells treated with or without MG132 and probed with indicated antibodies.

C  Left panel, Western blotting with HeLa and TRIM16$^{KO}$ cell lysates probed with indicated antibodies. Right panel, densitometric analysis (mean ± SD) of protein band intensity relative to actin, $n$ = 3, ***$P$ < 0.0005 (Student's unpaired $t$-test).

D–G  (D, F) HEK293T cells were either transiently transfected with an empty vector or a Flag-tagged-TRIM16 expressing vector and immunoblotted with Flag or indicated antibodies. L.E, low exposure; H.E, high exposure. (E, G) Densitometric analysis (mean ± SD) of protein band intensity relative to actin, $n$ = 3, *$P$ < 0.05 (Student's unpaired $t$-test).

H  WB analysis of HeLa and TRIM16$^{KO}$ lysates of cells treated with cycloheximide (100 μg/ml) for the indicated period and probed with different antibodies as indicated. The blots of control and TRIM16$^{KO}$ cells are exposed for the equal duration and developed together on the same X-ray film.

I  Quantification of NRF2, p62, and KEAP1 band intensities relative to actin. $n$ = 3, **$P$ < 0.005, *$P$ < 0.05 (Student's unpaired $t$-test).

J  Immunoprecipitation (IP) analysis of interaction between endogenous TRIM16 and endogenous p62 in HeLa cell lysates in absence and presence of MG132 (10 μM, 4 h). *The panel (J) here, Fig EV2H, and Appendix Fig S3H are part of same blots; hence, the input for TRIM16 is same.

K  The domain organization map of TRIM16 and deletion constructs cloned as Myc-tagged proteins.

L, M  IP analysis to map the interaction between p62 and TRIM16 domains in HEK293T cell lysates in (L) absence and (M) presence of MG132 (10 μM, 4 h).

N  IP analysis of the interaction between endogenous TRIM16 and endogenous NRF2 in HeLa cell lysates. M$^r$, molecular weight marker.

O  Co-IP analysis to map the interaction between NRF2 and TRIM16 domains in HEK293T cell lysates.

P  IP analysis of the interaction between endogenous TRIM16 and endogenous KEAP1 in HeLa cell lysates.

Source data are available online for this figure.

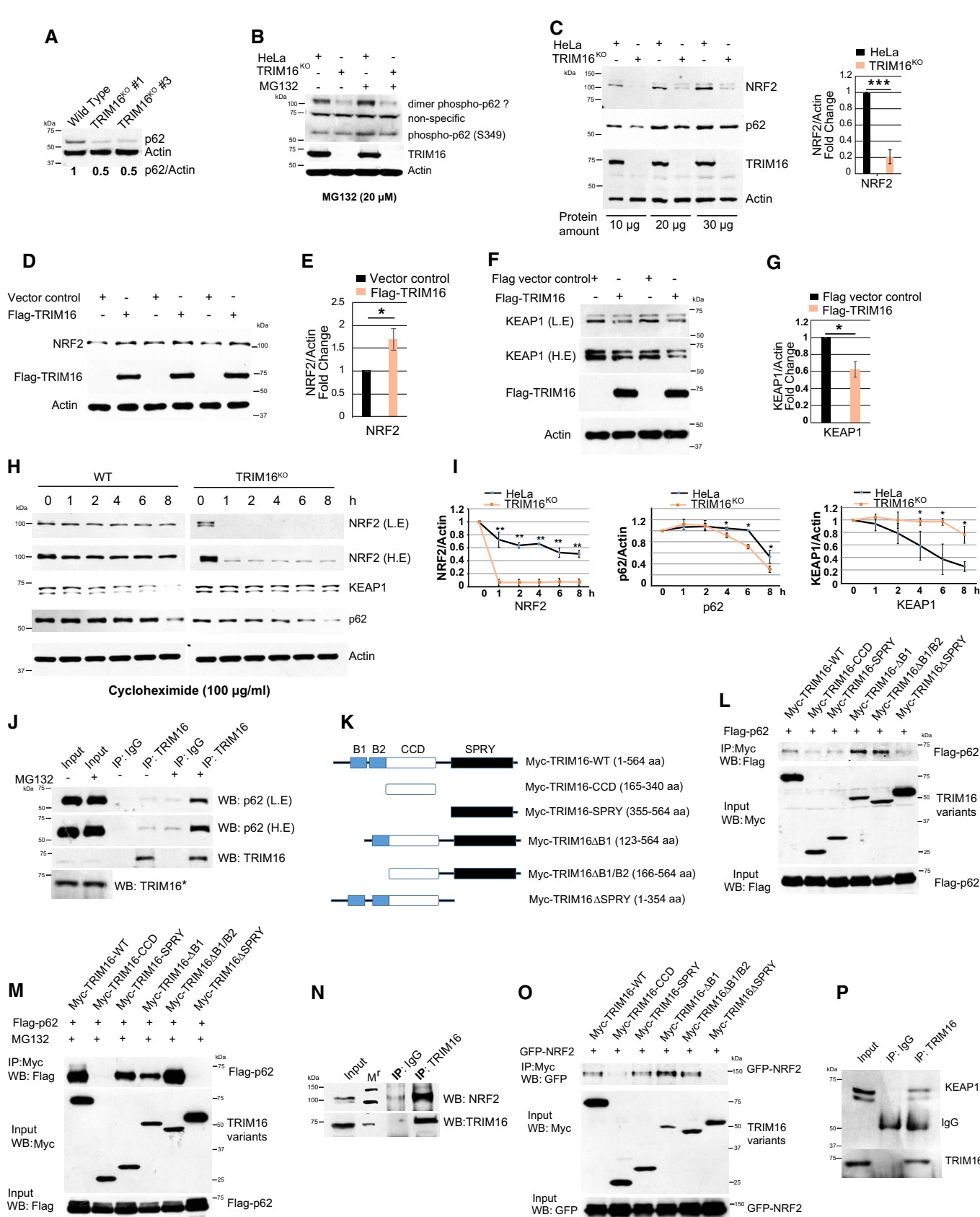

**Figure 2.**

increased p62 levels (Fig EV1A) suggesting that TRIM16 positively regulates p62 expression. Phosphorylation of p62 at serine 349 (in human) is indispensable for the p62-mediated NRF2-activation (Ichimura et al, 2013). The phosphorylation of p62 (S349) is increased on the treatment with MG132 in control cells but not in TRIM16$^{KO}$ cells indicating that TRIM16 is important for phosphorylation of p62 at S349 (Fig 2B). We next determined the levels of NRF2 in cells with altered expression of TRIM16. NRF2 levels (cytoplasmic and nuclear) were substantially less in TRIM16$^{KO}$ cells compared to the control cells (Figs 2C and EV2B), whereas overexpression of TRIM16 significantly increased the NRF2 levels (Fig 2D and E) indicating that TRIM16 positively regulates NRF2 expression too. On the contrary, KEAP1 expression was high in TRIM16-depleted cells, and its levels were decreased on overexpression of TRIM16 (Fig 2F and G, Appendix Fig S2B and C) suggesting a negative regulation.

Next, we asked whether TRIM16 regulates p62, NRF2, and KEAP1 expression at mRNA or protein levels. No significant change in the mRNA levels of Nrf2 and Keap1 was observed in TRIM16$^{KO}$ cells in comparison with control cells, whereas a small but significant reduction was detected in p62 mRNA expression (Appendix Fig S2D). Next, we performed cycloheximide chase experiments to explore the role of TRIM16 in stabilizing NRF2, p62 and KEAP1 proteins. Both TRIM16 depletion and the overexpression experiments show that TRIM16 stabilizes NRF2 and p62 whereas destabilizes the KEAP1 (Figs 2H and I, and EV2C–E). Altogether, the results suggest that TRIM16 regulates NRF2 and KEAP1 at the protein level, whereas it regulates p62 at both protein and mRNA levels.

Next, we investigated whether TRIM16 interacts and is part of p62, NRF2, and KEAP1 complex. The p62 interacts with TRIM16 weakly under basal conditions, and the interaction was increased significantly under the proteotoxic stress conditions (Fig 2J). These data are in agreement with the previous study where TRIM16-p62 direct interaction was found to be weak in in vitro conditions (Mandell et al, 2014a). This increased interaction may take place over the aggresomes where TRIM16 and p62 strongly co-localizes (see below, Fig 7E). Next, we determined the specificity of TRIM16-p62 interaction by mapping the domains of TRIM16 required for the interaction with p62 under both, basal and proteotoxic stress conditions. TRIM16 contains two B box, one CCD, and one SPRY domain/s (Fig 2K). TRIM16 B box domains are shown to be important for its E3 ligase activity and CCD domain help in homodimerization or heterodimerization with other TRIM proteins (Bell et al, 2012). The SPRY domain of TRIM proteins is known to mediate protein–protein interactions (Marin, 2012; D'Cruz et al, 2013). Under basal or proteotoxic stress conditions, the SPRY domain deleted TRIM16 protein (containing B1/B2 box and CCD domain) was not able to interact with p62 suggesting that SPRY domain is required for TRIM16-p62 interaction (Fig 2L and M). In agreement with this notion, the SPRY domain alone can interact with p62 (Fig 2L and M). The B1/B2 box deleted TRIM16 (containing SPRY and CCD domain) interacted with p62 stronger than wild-type protein, suggesting that B1/B2 boxes may hinder the TRIM16-p62 interaction (Fig 2L and M). This could be a regulatory mechanism to avoid unspecific interaction in the absence of stimulation.

Endogenous NRF2 strongly interacted with TRIM16 under basal conditions (Fig 2N), and this interaction was lost under proteotoxic stress conditions (Appendix Fig S2E). The MG132 treatment

(proteotoxic stress) is known to cause translocation of NRF2 to the nucleus (Sahni et al, 2008; Cui et al, 2013). However, TRIM16 remains cytoplasmic in these conditions (refer to Fig 7E). This may provide an explanation for the loss of TRIM16-NRF2 interaction after MG132 treatment. Co-immunoprecipitation assays with overexpressed proteins also revealed a strong interaction between TRIM16 and NRF2 (Fig EV2F, Appendix Fig S2F). Next, we mapped domain/s of TRIM16 required for interaction with NRF2. The deletion of SPRY domain completely abolished the interaction between TRIM16 and NRF2 (Fig 2O). Also, the TRIM16 SPRY domain alone can efficiently interact with NRF2 suggesting that like in case of p62, SPRY domain of TRIM16 is crucial for its interaction with NRF2 (Fig 2O). We found that TRIM16 also associates with KEAP1 (Figs 2P and EV2G) and this interaction was unchanged under proteotoxic stress conditions (Fig EV2H).

Next, we sought to determine the reason for the TRIM16-mediated stability of NRF2. As described above, KEAP1 is a negative regulator of NRF2 stability. We found that TRIM16 displaces KEAP1 from NRF2 (Fig 3A and B). The p62 on the other hand increases the stability of NRF2. We found that TRIM16 enhances the p62-NRF2 interaction (Fig 3C). The p62 interacts with KEAP1, sequesters it, and mediates its autophagic degradation. TRIM16 was found to increase the interaction between KEAP1 and p62 in immunoprecipitation assays (Fig 3D). In our model, TRIM16 stabilizes NRF2 by utilizing at least two mechanisms, one, by displacing KEAP1, and two, by increasing p62-NRF2 interaction (Fig 3E). Further, TRIM16 enhances the KEAP1 interaction with p62 leading to the sequestration and autophagic degradation of KEAP1 (Fig 3E).

Taken together, the data shown here suggest that TRIM16 is an integral part of the p62-KEAP1-NRF2 system and positively regulates expression of p62 and NRF2.

## TRIM16 affects ubiquitination status of NRF2 and modulates its activity

TRIM16 belongs to TRIM family proteins whose most of the members are RING domain containing E3 ligases (Hatakeyama, 2017). TRIM16 is devoid of RING domain; however, it still holds E3 ligase activity (Bell et al, 2012). So next we tested whether TRIM16 affects ubiquitination status of NRF2. The K48-linked poly-ubiquitination of a protein is known to target it for proteasomal degradation and the K63-linked poly-ubiquitination is important for varied functions including protein trafficking, proteins oligomerization, protein stability, and activation of signaling pathways (Jiang et al, 2012; Nazio et al, 2013; Erpapazoglou et al, 2014; Choi et al, 2016). To determine how TRIM16 affects NRF2 ubiquitination, we performed immunoprecipitation assays in the presence of two different variants of ubiquitin proteins, one that can only be ubiquitinated at lysine 48 residue (HA-K48-UB) and other that can only be ubiquitinated at lysine 63 residue (HA-K63-UB). Interestingly, we found that TRIM16 decreased K48-linked poly-ubiquitination and increased K63-linked poly-ubiquitination of NRF2 (Fig 4A–C). In similar experiments, no change in NRF2 ubiquitination was observed in the presence of GFP control vector (Appendix Fig S3A and B). To further corroborate the finding, we replaced HA-Ub-K63 with histidine-tagged ubiquitin-K63 (all lysine mutated except K63) and performed Ni-NTA pull-down assays in denaturing (6M guanidine–HCl) and non-denaturing conditions. The data show that TRIM16 increases conjugation of

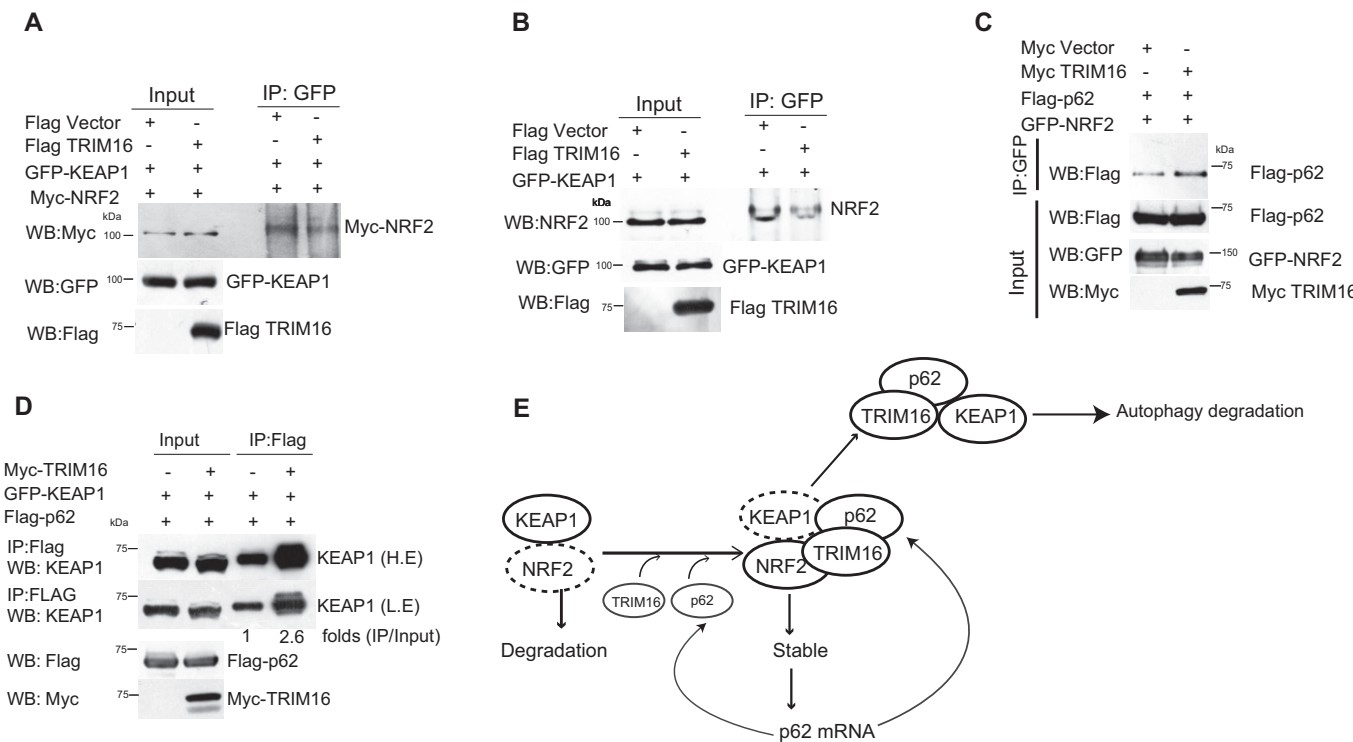

**Figure 3.    TRIM16 perturbs the interactions of the p62-KEAP1-NRF2 complex.**

A, B    Co-IP analysis of the interaction between NRF2 and KEAP1 in absence and presence of TRIM16.
C    Co-IP analysis of the interaction between NRF2 and p62 in absence and presence of TRIM16.
D    Co-IP analysis of the interaction between KEAP1 and p62 in absence and presence of TRIM16.
E    Graphical representation of data presented in Figs 2 and 3.

Source data are available online for this figure.

ubiquitins with NRF2 (Fig 4D and E). The decrease in K48-linked ubiquitination of NRF2 could be due to the TRIM16-mediated displacement and degradation of KEAP1 which along with Cullin3 is known to cause K48-linked ubiquitination and degradation of NRF2 (Kansanen *et al*, 2013). Several of the proteins get stabilized upon K63-linked ubiquitination (Nazio *et al*, 2013; Choi *et al*, 2016). Thus, both of these events can provide reasons for enhanced stability of NRF2 in the presence of TRIM16.

Next, we examined that which domain/s of TRIM16 is/are involved in K63-linked ubiquitination of NRF2 (Fig 4F). The data show that B1/B2 domains which were previously shown to be important for E3 ligase activity of TRIM16 (Bell *et al*, 2012) play an important role in K63-linked ubiquitination of NRF2 as their deletion substantially reduced the K63-linked ubiquitination of NRF2 (Fig 4F). The SPRY domain also appears to play a role in K63-linked ubiquitination of NRF2, as the ubiquitination was low in the absence of the SPRY domain and also SPRY domain itself is sufficient to increase K63-linked ubiquitination of NRF2 (Fig 4F). It appears that either SPRY domain itself can ubiquitinate NRF2 or it interacts with some other E3 ligase/s to achieve this. The latter is more likely, as the TRIM proteins are known for their tendency to heterodimerize with other TRIM proteins (Bell *et al*, 2012; Marin, 2012; Hatakeyama, 2017).

We found that the SPRY domain of TRIM16 is required for its interaction with p62 and NRF2 (Fig 2L and O). Further, along

with B1/B2 boxes, SPRY domain increases K63-linked ubiquitination of NRF2 (Fig 4F). We next complemented the TRIM16[KO] cells with TRIM16 deletion constructs to understand the role of TRIM16 domains in the stability of NRF2 and p62 under proteotoxic stress conditions (Fig 4G). The full-length TRIM16 is able to completely restore the NRF2 and p62 levels in TRIM16[KO] cells, again showing the role of TRIM16 in maintaining the stability of NRF2 and p62. Notably, the SPRY domain alone can completely restore NRF2/p62 levels (Fig 4G). Also, the SPRY domain deletion construct was defective in replenishing the NRF2/p62 expression suggesting that SPRY domain is required for full expression of NRF2/p62. The B1/B2 boxes mutant also appear to be slightly defective in the restoration of NRF2/p62 levels. Taken together, the data indicate that the SPRY domain of TRIM16 by its ability to mediate TRIM16 interaction and K63-linked ubiquitination of NRF2 plays a critical role in NRF2/p62 stability and expression.

To understand the functional significance of the results obtained above, we complemented the TRIM16[KO] cells with full-length or TRIM16 deletion constructs and assayed for protein aggregate formation under proteotoxic stress conditions (Fig EV3A and B). The results show that full-length myc-TRIM16 construct nicely restored the defective aggregate formation capacity of TRIM16[KO] cells (Fig EV3A and B). However, TRIM16 constructs deleted of B1/B2 or SPRY domains cannot (Fig EV3A and B). Interestingly, the

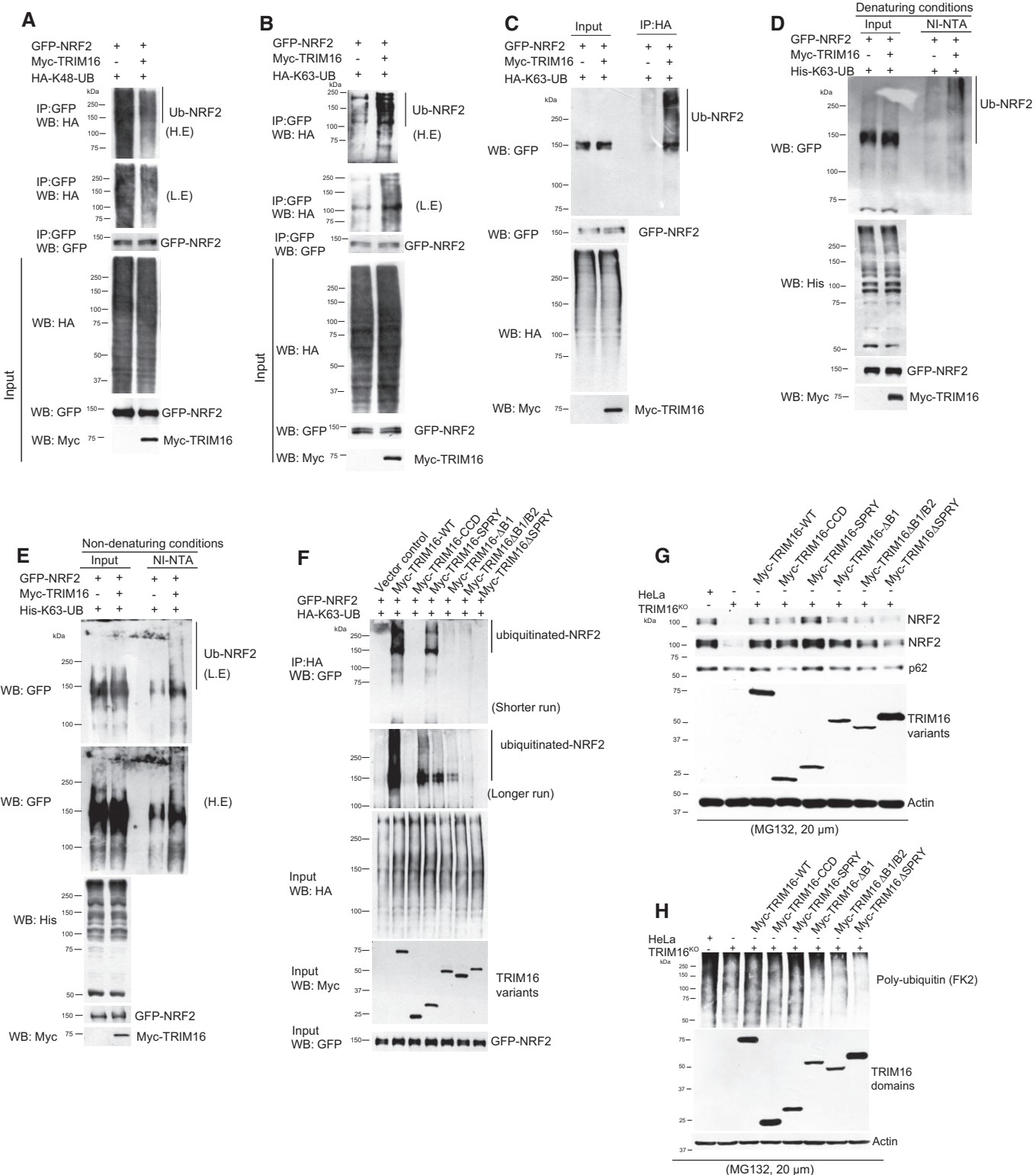

**Figure 4.**

TRIM16 SPRY domain can complement the protein aggregate formation capacity of TRIM16^KO cells, whereas CCD domain cannot. Similar results were obtained in the Western blotting experiment with the poly-ubiquitin antibody (Fig 4H). These data are in agreement with the important role of TRIM16 SPRY and B1/B2 domains in interaction, ubiquitination, and stability of p62 and NRF2. Both of the proteins are absolutely required for protein aggregate formation.

**Figure 4.  TRIM16 regulates ubiquitination of NRF2.**

A–C   Analysis of NRF2 ubiquitination in absence and presence of TRIM16 by co-IP assays using transiently transfected plasmid constructs as indicated. Two different variants of ubiquitin protein are used, one that can only be ubiquitinated at lysine 48 residue (HA-K48-UB) and other that can be only ubiquitinated at lysine 63 residue (HA-K63-UB). All other lysine residues are mutated. L.E, low exposure; H.E, high exposure.

D, E   Analysis of NRF2 ubiquitination in absence and presence of TRIM16 by Ni-NTA pull-down assays using transiently transfected plasmid constructs as indicated. His-tagged ubiquitin which is mutated at all lysines except 63 position is used in these assays.

F   Analysis of NRF2 ubiquitination in absence and presence of TRIM16 deletion variants by co-IP assays using transiently transfected plasmid constructs as indicated.

G   Western blot analysis of lysates from HeLa, TRIM16[KO] cells, and TRIM16[KO] cells complemented with TRIM16 deletion constructs where cells were treated with MG132 (20 μM, 2 h) and the blot is probed with indicated antibodies.

H   Western blot analysis of lysates from HeLa, TRIM16[KO] cells, and TRIM16[KO] cells complemented with TRIM16 deletion constructs where cells were treated with MG132 (20 μM, 2 h) and the blot is probed with indicated antibodies.

Source data are available online for this figure.

## TRIM16 controls NRF2 stress response and ubiquitination pathway gene expression

NRF2 is a master transcription regulator of the antioxidant and anti-proteotoxic stress response (Takaya *et al*, 2012; Jaramillo & Zhang, 2013; Kansanen *et al*, 2013). Next, we investigated the TRIM16-mediated regulation of NRF2 under stress conditions. Oxidative stress ($H_2O_2$ treatment) induces concentration- and time-dependent expression of NRF2, p62, and TRIM16 in control cells (Fig 5A, Appendix Fig S3C). However, under similar conditions, a rapid reduction in NRF2 and p62 was observed in TRIM16-depleted cells (Fig 5A, Appendix Fig S3C) suggesting that oxidative stress-induced expression or the stability of NRF2 and p62 is dependent on TRIM16. This notion is further supported by an elegant experiment where we chased expression of NRF2, p62, and TRIM16 at higher concentration of $H_2O_2$ (400 μM) for different time points (Fig 5B). In control cells, the expression of NRF2, p62, and TRIM16 was increased in 30 min and after that it rapidly decreased till 2 h time point (Fig 5B and C). After this time point, we again pulsed the cells with $H_2O_2$ (400 μM, red arrow in the Fig 5B and C), which again increased the expression of all the three proteins and thereafter protein levels rapidly dropped at the 6 h time point (Fig 5B and C). Of note, in TRIM16-depleted cells, this cyclic expression was not observed and the NRF2 and p62 levels were constantly and rapidly reduced (Fig 5B and C). These data suggest that under oxidative stress conditions, the expressions of TRIM16, NRF2, and p62 are co-regulated and TRIM16 is essential for increased expression and

stability of NRF2 and p62. The MG132 treatment induces TRIM16-dependent expression of NRF2 (Fig EV3C and D); however, p62 protein expression (but not mRNA, see below) was reduced in the presence of MG132 (Fig EV3D). The latter could be due to auto-phagy inducing property of MG132 which is known to degrade p62 (Fig EV3D, bafilomycin A1 lanes; Lan *et al*, 2015; Bao *et al*, 2016). $As_2O_3$, another potent oxidative stress inducer, increases the expression of NRF2 and p62, and ubiquitin conjugation of proteins (Fig 5D). This effect was abolished in the absence of TRIM16 (Fig 5D). Overall, the results suggest a pivotal role of TRIM16 in oxidative/proteotoxic stress-induced expression of NRF2 and p62.

The induction of heme oxygenase-1 (HO-1), NAD(P)H dehydrogenase (quinone) 1 (NQO1), and p62 is a hallmark of the NRF2-mediated stress response (Loboda *et al*, 2016). The MG132 induced mRNA expression of *p62*, *Nqo1*, and *Hmox1* (Appendix Fig S3D) was significantly reduced in TRIM16-depleted cells (Fig 5E–G). Further, just overexpression of TRIM16 increased the mRNA levels of *p62*, *Nqo1*, and *Hmox1* and this increased expression were blunted on knocking down NRF2 (Fig 5H–J). Taken together, the data suggest that TRIM16 is required for NRF2-mediated stress response.

Proteasomal dysfunction and oxidative stress induce imbalance in protein homeostasis and lead to increased Ub-tagged misfolded proteins which subsequently form protein aggregates. $H_2O_2$ and MG132 treatments induce poly-ubiquitination of misfolded proteins in control cells (Figs 5K and EV3E and F). In contrast, there was marked reduction in poly-ubiquitination of proteins in TRIM16[KO]

**Figure 5.  TRIM16 regulates stress-induced NRF2 response including ubiquitination of protein aggregates.**

A   Western blot analysis of HeLa and TRIM16[KO] lysates of cells treated with increasing concentrations of $H_2O_2$ as specified and probed with antibodies as indicated.

B   WB analysis of HeLa and TRIM16[KO] lysates of cells treated with 400 μM of $H_2O_2$ for different durations as indicated and probed with antibodies as shown. The red arrows indicate the time point where the cells were retreated with 400 μM of $H_2O_2$. L.E, low exposure; H.E, high exposure.

C   Densitometric analysis of NRF2 band intensity (in panel B) normalized with actin.

D   WB analysis of HeLa and TRIM16[KO] lysates of cells treated with 2.5 μM (2 h) of $As_2O_3$.

E–G   RNA isolated from HeLa and TRIM16[KO] cells, untreated or treated with MG132 (20 μM, 2 h), were subjected to qRT–PCR with primers of genes as indicated. The fold induction in MG132-treated samples is calculated relative to untreated samples. Mean ± SD, *n* = 3, *$P$ < 0.05, **$P$ < 0.005 (Student's unpaired *t*-test).

H–J   RNA isolated from control and NRF2 knockdown HeLa cells transfected with control vector or Myc-TRIM16 subjected to qRT–PCR with primers of genes as indicated. Mean ± SD, *n* = 3, **$P$ < 0.005, ***$P$ < 0.0005 (Student's unpaired *t*-test).

K   Western blot analysis of HeLa and TRIM16[KO] lysates of cells treated with h increasing concentrations of $H_2O_2$ as specified and probed with antibodies as indicated.

L   Densitometric analysis of Ub smear intensity normalized with actin. Mean ± SD, *n* = 3, *$P$ < 0.05 (Student's unpaired *t*-test).

M   Representative confocal images of control, *Ubb*, and *Ube2n* siRNA transfected cells treated with $H_2O_2$ (200 μM, 2 h) and IF analysis was performed with Ub and p62 antibodies. Scale bar: 10 μm.

N   The graph shows the percentage of cells with protein aggregates. Data from ≥ 10 microscopic fields for each condition (40×), *n* = 2, mean ± SD.

O   Western blot analysis of *Ubb* and *Ube2n* siRNA transfected cell lysates with antibodies as indicated.

P   Pictorial representation of results obtained in Figs 1–5.

Source data are available online for this figure.

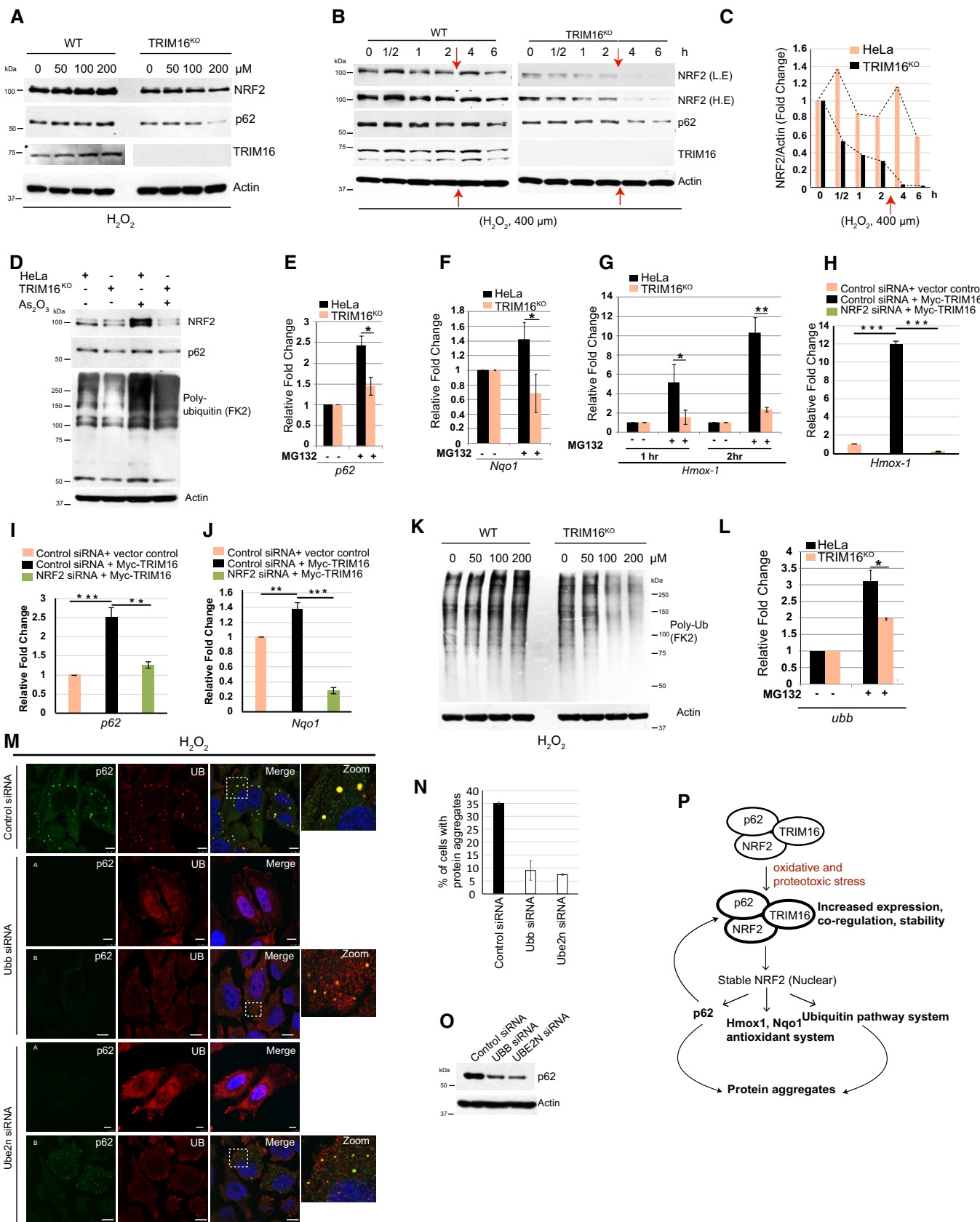

Figure 5.

cells (Figs 5K and EV3E and F) indicating that TRIM16 is important for poly-ubiquitination of misfolded proteins formed during oxidative or proteotoxic stress conditions. Both K48-linked and K63-linked ubiquitination of proteins were reduced in the absence of TRIM16 (Fig EV3G). We hypothesize that TRIM16 via NRF2 upregulates expression of genes required for ubiquitination of misfolded proteins. In mammals, the *Ubb* (ubiquitin B) and *Ubc* (ubiquitin C) genes encode for poly-ubiquitin precursor proteins and are essential for poly-ubiquitination of misfolded proteins (Pankiv *et al*, 2007; Lim *et al*, 2015; Pajares *et al*, 2017). Transcription of both the *Ubb* and *Ubc* genes was increased upon treatment of cells with MG132 (Appendix Fig S3D). The MG132-induced *Ubb* but not *Ubc* transcription was significantly attenuated in TRIM16[KO] cells (Figs 5L and EV3H). Next, we examined the expression of several other sentinel ubiquitin pathway genes, including *Uba-1* (ubiquitin-activating enzyme, E1), *Uba-3* (ubiquitin-activating enzyme, E1), *Ube2h* (ubiquitin-conjugating enzyme, E2), *Ube2n* (ubiquitin-conjugating enzyme, E2), *Ube2z* (ubiquitin-conjugating enzyme, E2), *Ubr2* (ubiquitin-protein ligase, E3), *Cullin4B* (ubiquitin-protein ligase, E3), and *Nedd8* (a ubiquitin-like protein). The expression of E1 enzymes, *Uba-1* but not *Uba-3,* was induced by MG132, and this increase was significantly attenuated in TRIM16[KO] cells (Fig EV3I). The transcription of all the three E2 enzymes (*UBe2h, Ube2n,* and *Ube2z*) tested was upregulated by MG132 and was reduced in TRIM16[KO] cells (Fig EV3J). The *Ubr2* and *Cullin4b* transcriptions were increased, but *Nedd8* transcription was unaffected by MG132 (Appendix Fig S3E). *Ubr2* and *Cullin4b* expressions were unaffected in TRIM16[KO] cells (Appendix Fig S3F). To confirm the role of these ubiquitin genes in protein aggregate biogenesis, we knocked down two of the TRIM16-regulated ubiquitin pathway genes, *Ubb,* and *Ube2n* and measured the accumulation of protein aggregates under oxidative stress conditions. Very clearly, both of the genes are required for protein aggregate formation (Fig 5M and N). In very few of the *Ubb* and *Ube2n* knockdown cells, we observed the formation of very small protein aggregates (B panels, Fig 5M) suggesting that both of the gene products are required for proper and full assembly of protein aggregates. One of the reasons for reduced protein aggregates could be the reduced expression of p62 upon depletion of *Ubb* and *Ube2n* (Fig 5O). These data show that TRIM16 is important for proteotoxic/oxidative stress-induced transcription of several sentinel genes of ubiquitin system (*Ubb, Uba-1, Ube2h,*

*Ube2n,* and *Ube2z*) and is required for ubiquitination of misfolded proteins leading to the formation of protein aggregates.

Taken together, we show that under stress conditions, TRIM16 co-regulates with NRF2 and p62 and is required for stress-induced expression and stability of NRF2 and p62 (Fig 5P). The TRIM16 via stabilized NRF2 induces p62, ubiquitin system pathway genes, and antioxidant genes for stress response which includes biogenesis of protein aggregates (Fig 5P).

### In a feedback loop, NRF2 regulates TRIM16 expression

Previous studies indicate, however, without any direct evidence that TRIM16 is an NRF2-regulated stress-induced gene (Noguchi, 2008; Python *et al*, 2009; Hirose *et al*, 2011; Dodson *et al*, 2015; Lacher & Slattery, 2016; Pajares *et al*, 2017). We show that NRF2 silencing by siRNA results in decreased TRIM16 expression both at the protein and at the mRNA level (Figs 6A and EV4A and B). The p62 which is a known NRF2-targeted gene also showed reduced expression (Figs 6A and EV4A and B). So NRF2 positively regulates both TRIM16 and p62 expression, and both TRIM16 and p62 are the positive regulators of NRF2 resulting in a regulatory feedback loop.

### NRF2 is a central player in stress-induced TRIM16 protein aggregate biogenesis response

Given the role of NRF2 in stress-induced detoxification processes, we hypothesize that some of the TRIM16 functions revealed above are mediated through NRF2. Indeed, the above-mentioned ubiquitin system genes are regulated by NRF2 (Figs 6B and C, and EV4B) indicating that TRIM16 modulates the expression of these genes through NRF2-mediated transcriptional regulation. Further, like TRIM16, the knockdown of NRF2 reduced the MG132- and H$_2$O$_2$-induced poly-ubiquitination of misfolded proteins or protein aggregates (Figs 6D and E, and EV4C). Furthermore, we found that NRF2 depletion substantially reduced the Ub, p62, and LC3B marked protein aggregates formed during oxidative and proteotoxic stress conditions (Figs 6F and G, and EV4D and E). This phenotype of NRF2 depletion is noticeable and maybe because of the reduced expression of TRIM16, p62, and ubiquitin genes in these cells (Fig 6A and B).

Next, we examined, whether NRF2 complementation in TRIM16[KO] cells can compensate for the ubiquitination and protein

**Figure 6. NRF2 regulates TRIM16, ubiquitin pathway genes expression, and biogenesis of protein aggregates.**

A       Control or NRF2 siRNA transfected HeLa cells lysates were subjected to Western blotting with indicated antibodies.

B, C    RNA isolated from untreated or MG132-treated (20 μM, 2 h) control or NRF2 siRNA transfected cells was subjected to qRT–PCR with primers of genes as indicated. The fold induction in MG132-treated samples is calculated relative to untreated samples. Mean ± SD, *n* = 3, *P < 0.05 (Student's unpaired *t*-test).

D, E    WB analysis of control siRNA or NRF2 siRNA transfected lysates of HeLa cells, untreated or treated with MG132 (20 μM, 2 h) or H$_2$O$_2$ (200 μM, 2 h) as indicated and probed with antibodies as shown. L.E, low exposure; H.E, High Exposure.

F, G    Left panels: Representative confocal images of control siRNA and NRF2 siRNA transfected cells treated with (F) MG132 (20 μM, 2 h), (G) H$_2$O$_2$ (200 μM, 2 h), where IF analysis was conducted with Ub and p62 antibodies. Right panels: Fluorescence intensity line is tracing corresponding to a white line in zoom panel.

H       HeLa cells, TRIM16[KO] cells, and TRIM16[KO] cells complemented with Myc-NRF2 or GFP-NRF2 were treated with MG132 (20 μM, 2 h) and subjected to WB analysis with Myc, GFP, and indicated antibodies.

I       Representative confocal images of HeLa cells, TRIM16[KO] cells, and TRIM16[KO] cells complemented with Myc-NRF2 which were treated with MG132 (20 μM, 2 h) and subjected to IF with Ub and p62 antibodies.

J       Graph represent % of cells with Ub-p62 dots, data from ≥ 10 microscopic fields (40×), *n* = 3, mean ± SD, *P < 0.005 (one-way ANOVA test).

K       Graphical representation of results obtained.

Data information: Unless otherwise stated, scale bar: 10 μm.
Source data are available online for this figure.

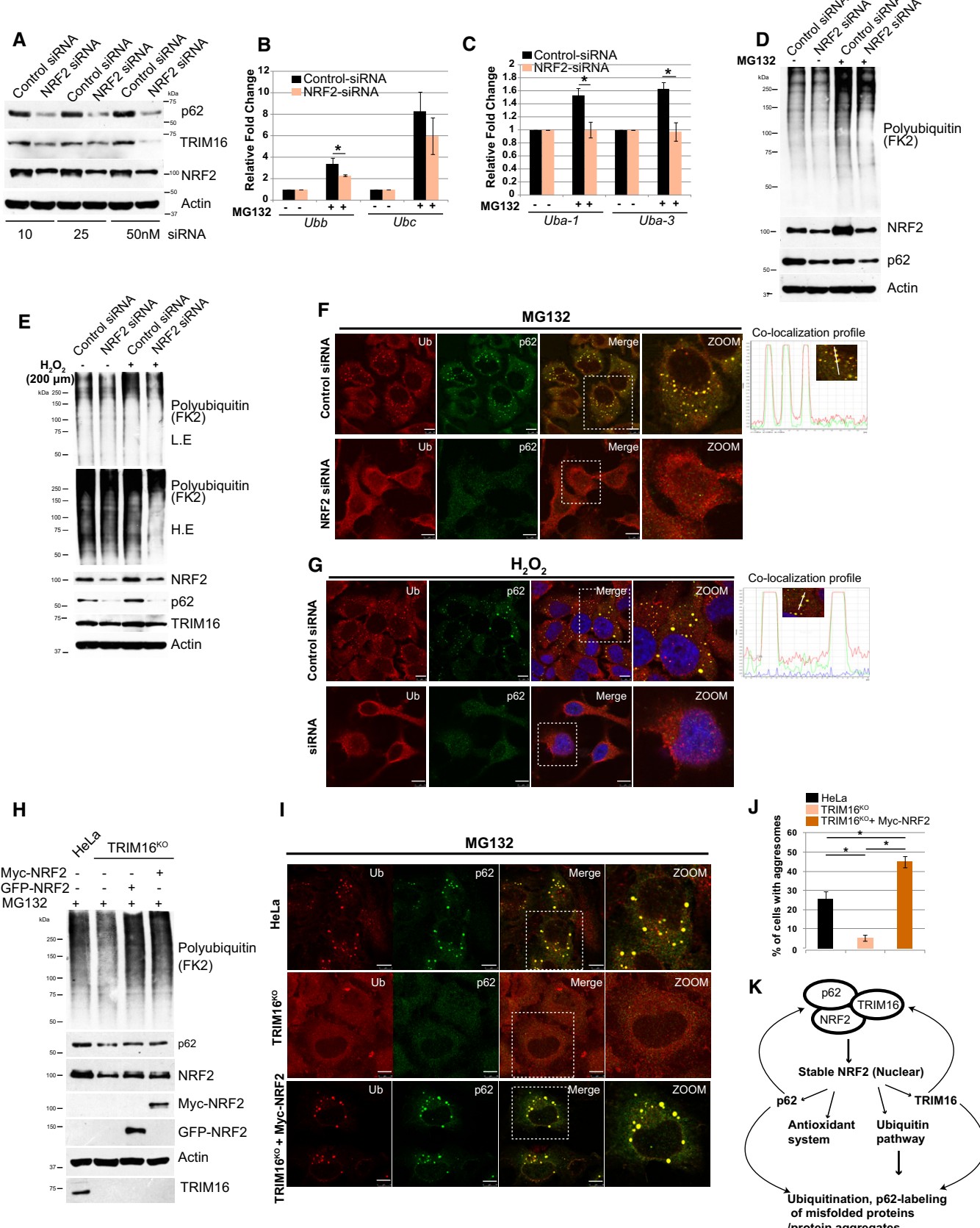

**Figure 6.**

aggregate formation defects observed in TRIM16-depleted cells. We found that the NRF2 complemented TRIM16$^{KO}$ cells showed an increased ability of protein ubiquitination (Fig 6H) and Ub-p62-positive aggresome formation (Figs 6I and J, and EV4F). The observed increased accumulation of Ub-p62-positive aggresomes in NRF2-complemented cells compared to wild-type cells could be due to their incapability of autophagic degradation in the absence of TRIM16 which is important for degradation of protein aggregates (see below). Altogether, we show that the NRF2 plays a central role in the TRIM16-mediated stress response of formation of protein aggregates from misfolded proteins (Fig 6K).

## TRIM16 act as a scaffold protein and is important for autophagic clearance of protein aggregates

TRIM16 is a positive regulator of autophagy (Figs 1K and L, and EV1I and J). Next, we tested whether TRIM16 in addition to biogenesis has a role in the autophagic degradation of protein aggregates. Since TRIM16 is required for ubiquitination and protein aggregation, it is not possible to study the role of TRIM16 in degradation of endogenous protein aggregates in isolation. To investigate this, we studied the role of TRIM16 in degradation of protein aggregates formed by self-oligomerization (such as p62) or aggregation-prone proteins (such as expanded huntingtin exon 1 Q74). Recent studies suggest that the ubiquitination process is not essential for the formation of aggregates of such proteins (Gal et al, 2009; Riley et al, 2010; Bersuker et al, 2016); however, it is important for their degradation via autophagy (Bjorkoy et al, 2005; Pankiv et al, 2007). Since self-oligomerizing or aggregation-prone proteins form aggregates independent of ubiquitination, their aggregation may not be hindered in TRIM16-depleted cells, and therefore, utilizing this system, we can study the role of TRIM16 in degradation of protein aggregates. We overexpressed p62 in control and TRIM16$^{KO}$ cells. Interestingly, we found that the p62 aggregates were accumulated more in TRIM16$^{KO}$ cells compared to the control cells in immunofluorescence and also in soluble/insoluble fractionation Western blotting experiments (Figs 7A and B, and EV5A and B) indicating the role of TRIM16 in degradation of p62 aggregates. The p62 aggregates are predominantly degraded by autophagy process (Bjorkoy et al, 2005). A similar increased accumulation of GFP–Huntington exon 1 (Q74) aggregates, another well-documented autophagy cargo (Ravikumar et al, 2004; Komatsu et al, 2007a), was observed in

insoluble fraction of TRIM16$^{KO}$ cells compared to HeLa cells (Fig 7C and D) underscoring the importance of TRIM16 in degradation of autophagic cargoes. Further, the p62 aggregates in TRIM16-depleted cells displayed reduced ubiquitination (Fig EV5C and D, Appendix Fig S3F) and reduced co-localization with LC3B (Fig EV5E and F). Taken together, these data indicate that TRIM16 is important for degradation of protein aggregates destined for autophagic degradation.

To further substantiate this observation, we determined whether TRIM16 is physically present over the aggresomes and participates directly in their turnover. Under basal conditions, the GFP-TRIM16 mostly showed diffused cytoplasmic staining or occasional dots (aggregates) co-localizing with Ub but not with p62 (Fig 7E and F). In MG132-treated cells, GFP-TRIM16 strongly co-localizes with Ub-positive aggresomes; however, the triple co-localization of TRIM16, Ub, and p62 was only observed in a few of the cells (Fig 7E and F). Since MG132 induces autophagy flux which can lead to the autophagic degradation of p62 and protein aggregates (Lan et al, 2015; Bao et al, 2016), we performed immunofluorescence analysis in cells treated with MG132 and autophagy flux inhibitor, bafilomycin A1. This treatment substantially increased the co-localization between TRIM16, Ub, and p62 (Fig 7E and F). A clear co-localization was also observed between TRIM16 and LC3B in the presence of MG132 or MG132 and bafilomycin A1 (Fig 7G). Further, in immunoprecipitation assays, the affinity of TRIM16 for p62 and LC3B was increased upon treatment of the cell with MG132 (Fig 2J, Appendix Fig S3G). The data suggest that TRIM16 is present over the aggresome and co-localizes with p62 and LC3B. To corroborate these findings, Western blotting experiment with soluble/insoluble fractions was performed. The TRIM16 was specifically accumulated in the insoluble (aggresomes containing) fractions of MG132-treated cells (Fig EV5G, compare lane 5 and 7). The presence of TRIM16 in protein aggregates and its requirement for the interaction of Ub, p62, and LC3B with the protein aggregates suggest that TRIM16 may act as a scaffold protein and assists in interactions. To assist in interactions, the scaffold proteins usually exist in oligomeric forms. Indeed, TRIM16 have tendency to self-oligomerize (Figs 7H and EV5H) and is present as high molecular weight oligomers in the protein aggregate fraction, along with p62 (Fig 7I). We further found that the CCD domain of TRIM16 was important for its oligomerization (Fig 7J). It was interesting to note that the p62 is required for aggregation of TRIM16 (Fig EV5I).

---

**Figure 7. TRIM16 mediates selective clearance of aggregated autophagy cargoes.**

A–D (A, C) WB analysis of detergent-soluble and -insoluble fractions of HeLa and TRIM16$^{KO}$ cells expressing either (A) Flag-p62 or (C) GFP-polyQ74 probed with Flag or GFP or indicated antibodies. L.E, low exposure; H.E, high exposure. (B, D) Densitometric analysis of WB's (shown in panels A and C) normalized with α-tubulin or lamin-B1.

E Confocal images of GFP-TRIM16-expressing HeLa cells, untreated or treated with MG132 (10 μM, 4 h) and/or bafilomycin A1 (300 nM, 3 h) and IF analysis is performed with antibodies as indicated. Right panel graphs: Fluorescence intensity line is tracing corresponding to a white line in zoom panel. Scale bar: 10 μm.

F The graph shows the percentage of cells with TRIM16-Ub or TRIM16-Ub-p62 co-localized dots/aggregates. Data from ≥ 10 microscopic fields, n = 2, mean ± SD.

G Representative confocal images of GFP-TRIM16 expressing HeLa cells, untreated or treated with MG132 (10 μM, 4 h) and/or bafilomycin A1 (300 nM, 3 h) and IF analysis is performed with antibodies as indicated. Right panel graphs: Fluorescence intensity line is tracing corresponding to a white line in zoom panel. Scale bar: 10 μm.

H Co-IP analysis of the interaction between two differently tagged TRIM16 variants.

I WB analysis of detergent-soluble and detergent-insoluble DSS cross-linked fractions of HeLa and TRIM16$^{KO}$ cells, untreated or treated with MG132 (10 μM, 4 h) and probed with indicated antibodies.

J Co-IP analysis of the interaction between Flag-tagged TRIM16 and Myc-tagged TRIM16 full length and its domain deletion proteins.

Source data are available online for this figure.

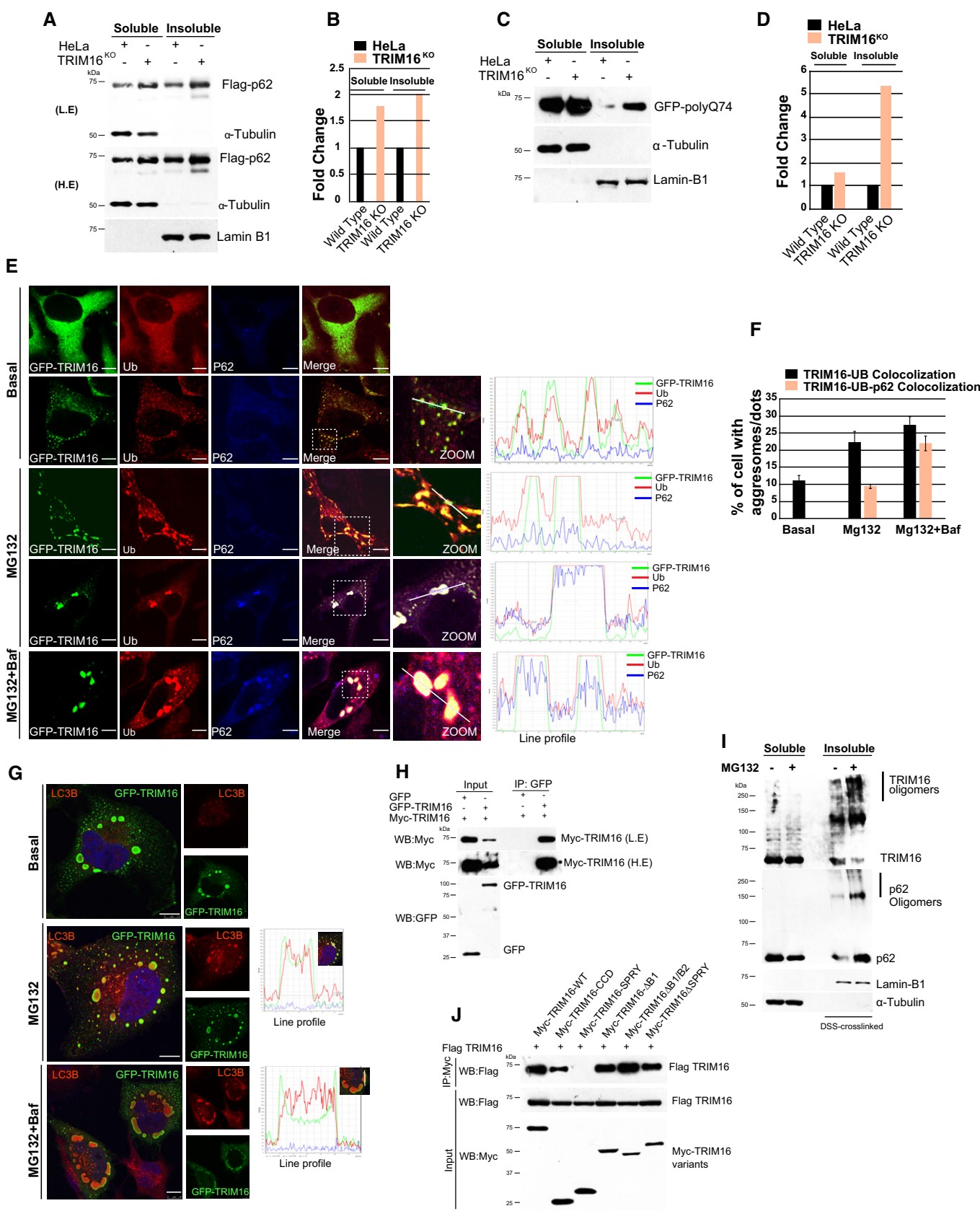

Figure 7.

We have shown in a previous study that TRIM16 interacts with ULK1 and ATG16L1, the autophagy initiation, and elongation protein, on the damaged endomembranes (Chauhan *et al*, 2016). We tested whether a similar association of TRIM16 and ULK1 or ATG16L1 takes place on the aggresomes. Under basal conditions, GFP-TRIM16 is localized diffusely in the cytoplasm, and a rare co-localization with either ULK1 or ATG16L1 was seen in HeLa cells, whereas in MG132-treated cells, TRIM16 nicely co-localized with ULK1 and ATG16L1 on Ub-positive aggresomes (Fig EV5J).

Overall, we show that TRIM16 is physically present on the protein aggregates and associates with the ubiquitin and autophagy, adaptor (p62), initiation (ULK1), and elongation (ATG16L1, LC3B) proteins to promote the autophagic degradation of the protein aggregates.

TRIM16 plays a role in the autophagic degradation of protein aggregates, but it appears that TRIM16 itself is not degraded by autophagy. On starving the cells, TRIM16 neither goes to lysosomal compartment as indicated by no change in color of dual-tagged TRIM16 (Fig EV5K) nor its protein levels were reduced (Appendix Fig S3H).

### TRIM16 protects cells from the toxic effects of oxidative and proteotoxic stresses

TRIM16 is important to mount an antioxidant and anti-proteotoxic stress response. We next investigated the role of TRIM16 in the survival of cells subjected to these stresses. The basal growth rate of TRIM16[KO] cells was comparable to the control cells (Fig 8A and B, Appendix Fig S3I). However, under oxidative and proteotoxic stress conditions, TRIM16[KO] cells were significantly growth defective (Fig 8A and B, Appendix Fig S3J and K). We next employed the Annexin-V/propidium iodide double-staining to measure the apoptosis and cell death under basal and stressed conditions. $As_2O_3$ and $H_2O_2$ were used as oxidative stress inducers, and puromycin and MG132 were used to induce proteotoxic stress. Under basal conditions, no significant difference in apoptosis was observed between TRIM16[KO] and control cells (Fig 8C–F). However, the TRIM16-depleted cells showed increased apoptosis and cell death on exposure to $As_2O_3$, $H_2O_2$, puromycin and MG132 (Fig 8C–F). Further, increased cleavage of caspase-3 and PARP-1, the apoptosis marker proteins were also observed in TRIM16-depleted cells on exposure of $As_2O_3$ or $H_2O_2$ (Fig 8G, Appendix Fig S3K). Taken together, TRIM16 prevents oxidative/proteotoxic stress-induced apoptosis/cell death.

### TRIM16 prevents oxidative stress-induced cytotoxicity *In vivo*

TRIM16-depleted cells showed reduced colony formation capacity under oxidative or proteotoxic stress conditions in clonogenic assays (Fig 9A and B). We employed xenograft BALB/C-nude mouse model to examine the role of TRIM16 in cell proliferation and growth *in vivo*. An equal number of control and TRIM16[KO] cells were injected subcutaneously in lower flanks of mice. $As_2O_3$ exposure is known to induce massive ROS production and DNA damage leading to cell death (Woo *et al*, 2002; Selvaraj *et al*, 2013). Once tumors are formed, a sub-lethal dose of $AS_2O_3$ (5 mg/kg) was injected intraperitoneally in mice every alternate day until the termination of the experiment (Fig 9C). In PBS (phosphate buffer saline)-treated control group, no substantial difference in proliferation capacity of HeLa vs. TRIM16[KO] cells was observed (Fig 9C and D). However, treatment of $As_2O_3$ rapidly regressed the TRIM16[KO] tumors without affecting the growth of control tumors (Fig 9C and D). No significant change in body weight of mice was observed during the course of the experiment (Appendix Fig S4A). The mice were sacrificed, and tumor weight was measured, which was significantly lower in $As_2O_3$-treated TRIM16[KO] group compared to other groups (Fig 9E and F, Appendix Fig S4B). Immunohistochemistry analysis (KI-67 staining) revealed reduced proliferation in TRIM16[KO] tumors compared to control tumors (Fig 9G). Western blot analysis of tumors lysates from two different mice emulated the key *in vitro* findings of this study that TRIM16 is important for NRF2 and p62 expression and is required for stress-induced poly-ubiquitination of misfolded proteins (Fig 9H). Altogether, we show in tumor model that TRIM16 modulates NRF2-p62 signaling and ubiquitin system to protect the cells against oxidative stress-induced toxicity.

## Discussion

The occurrence of misfolded proteins and formation of misfolded protein aggregates is an unavoidable consequence of cellular protein biogenesis. The assembly of misfolded protein aggregates is a cytoprotective measure resulting in the sequestration of highly toxic free misfolded proteins or intermediate aggregates (Arrasate *et al*, 2004; Tanaka *et al*, 2004; Douglas *et al*, 2008; Tyedmers *et al*, 2010). The misfolded protein aggregates are rapidly cleared to maintain cellular homeostasis. The unresolved accumulation of protein aggregates is cytotoxic and implicated in the pathogenesis of several diseases (Hara *et al*, 2006; Komatsu *et al*, 2007b; Polajnar & Zerovnik, 2014). The ubiquitin labeled misfolded proteins are recognized by p62 which drives their oligomerization into aggregates (Seibenhener *et al*, 2004; Bjorkoy *et al*, 2005; Clausen *et al*, 2010). Subsequently, p62 or other autophagy receptors present on the protein aggregates recruit LC3B and facilitate their autophagic degradation (Pankiv *et al*, 2007; Lamark & Johansen, 2012). However, our understanding

---

**Figure 8.  TRIM16 reduces cytotoxicity mediated by oxidative and proteotoxic stresses.**

A    MTT assays performed at different time points with HeLa and TRIM16[KO] cells, untreated or treated with $H_2O_2$ (400 μM, 2 h). Data, mean ± SD, n = 3, *P < 0.05 (Student's unpaired *t*-test).

B    Images of HeLa and TRIM16[KO] cells, untreated or treated with $As_2O_3$ (2.5 μM, 4 h). Scale bar: 400 μm.

C–F    Flow cytometry analysis of HeLa and TRIM16[KO] cells stained with Annexin-V/propidium iodide (double-staining), untreated or treated with (C) $As_2O_3$ (2.5 μM, 4 h), (D) $H_2O_2$ (400 μM, 2 h), (E) puromycin (5 μg/ml, 6 h), and (F) MG132 (20 μM, 8 h).

G    Immunoblot blot analysis of HeLa and TRIM16[KO] lysates of cells treated with 5 μM of $As_2O_3$ for different durations as indicated and probed with antibodies as shown. L.E, low exposure; H.E, high exposure. Arrowheads indicate the cleaved form of PARP-1 and caspase-3.

Source data are available online for this figure.

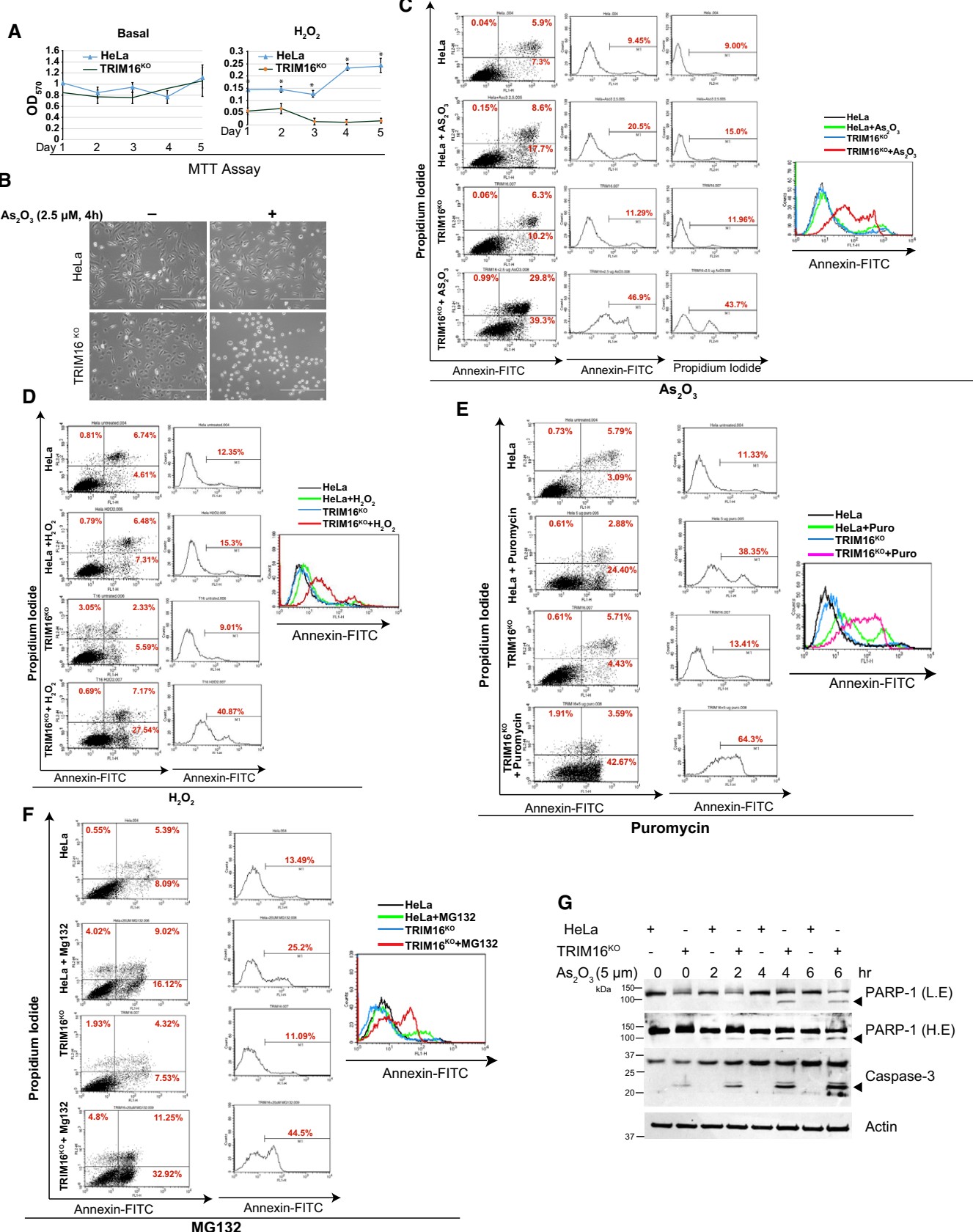

**Figure 8.**

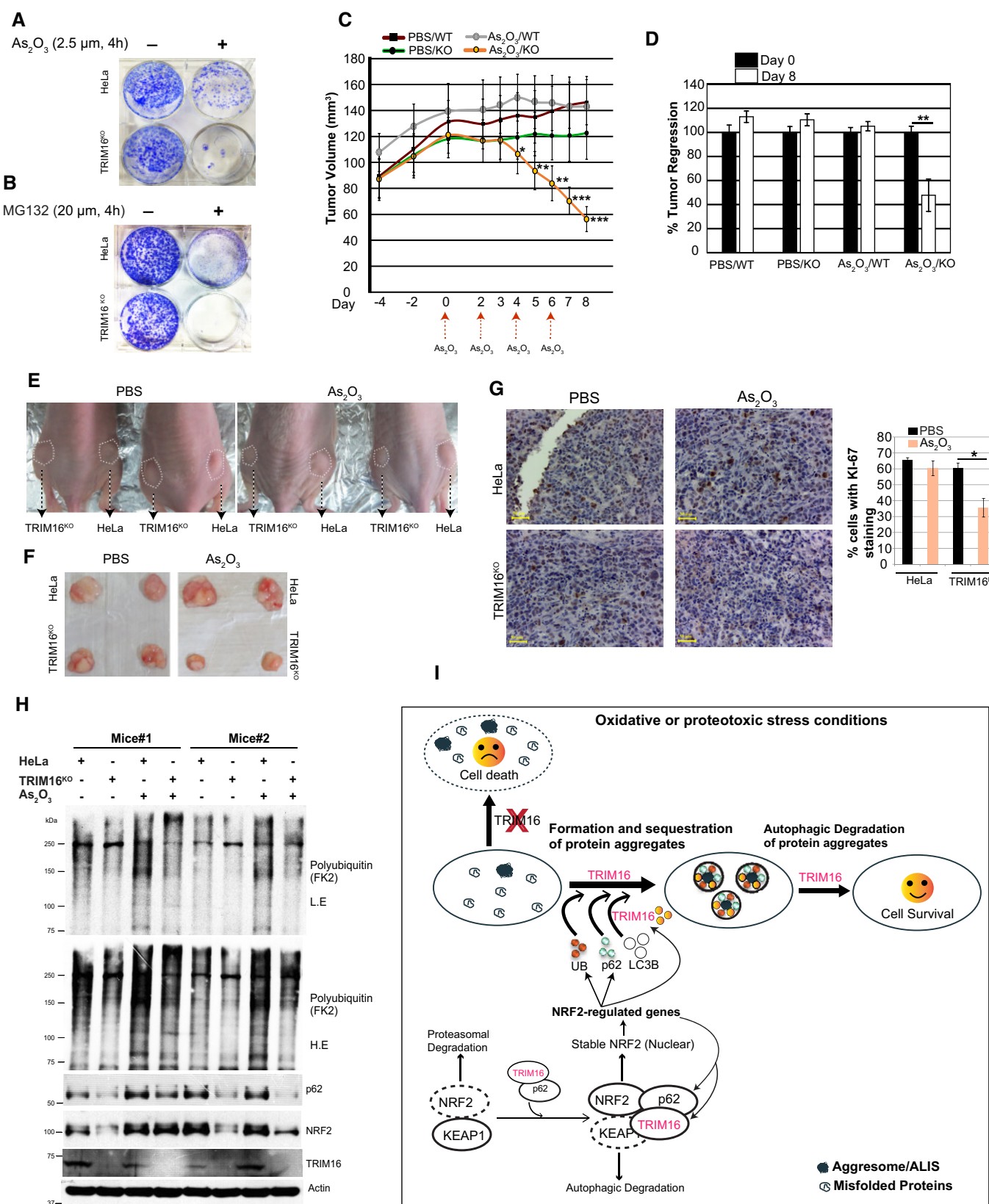

**Figure 9.**

**Figure 9.  TRIM16 is protective against oxidative stress-induced cytotoxicity *in vivo*.**

A, B  Clonogenic assays were performed with HeLa and TRIM16[KO] cells, untreated or treated with $As_2O_3$ or MG132.

C  Tumor volumes of HeLa and TRIM16[KO] tumors at indicated time points. $As_2O_3$ is injected when tumor volume of all groups was > 100 mm$^3$ (day 0 in the graph). Tumor volumes were measured on an interval as indicated. Mean ± SE, $n$ = 6 (each group), *$P$ < 0.05, **$P$ < 0.005, ***$P$ < 0.0005 (ANOVA).

D  Graph shows the % tumor regression (tumor volume on day 1 of $As_2O_3$ treatment/tumor volume on day of sacrifice × 100). Mean ± SE, $n$ = 6 (each group), **$P$ < 0.005 (Student's unpaired *t*-test).

E  Representative images of HeLa and TRIM16[KO] tumors formed in the nude mice in the absence and presence of $As_2O_3$.

F  Representative pictures of dissected tumors.

G  Immunohistochemistry analysis performed with KI-67 cell proliferation marker and hematoxylin nuclear stain. The graph represents the % of cells with KI-67 staining. > 400 cells were counted for this analysis from three different sections from different animals, mean ± SD, *$P$ < 0.05 (Student's unpaired *t*-test). Scale bar: 50 μm.

H  WB analysis of tumor tissue lysates from two different mice with antibodies as indicated. L.E, low exposure; H.E, high exposure.

I  Graphical representation of work presented in this study.

Source data are available online for this figure.

of this entire process is limited and the details of other molecular players that drive this process remain unknown.

In this study, we identified TRIM16 as a key regulatory protein that mounts a comprehensive response to oxidative or proteotoxic stresses and facilitates the biogenesis and degradation of the protein aggregates (Fig 9I). Through its two-pronged regulation of NRF2 and autophagy, TRIM16 serves as proteostasis sentinel during oxidative or proteotoxic stresses. TRIM16 acts as a scaffold protein for both biogenesis and degradation of protein aggregates. Mechanistically, we show that under oxidative and proteotoxic stress conditions, TRIM16 upregulates and induces NRF2 signaling leading to increased cellular availability of ubiquitin system proteins and p62 for the assembly of protein aggregates (Fig 9I). TRIM16 increases interaction between p62 and NRF2, most likely to activate NRF2. TRIM16 also enhances interaction between p62 and KEAP1 for sequestration and autophagic degradation of KEAP1. In addition, TRIM16 is present as oligomer on stress-induced protein aggregates and facilitates ubiquitin, p62 and LC3B recruitment to the aggregates. TRIM16 is important for basal and proteotoxic stress-induced autophagy. TRIM16 associates with ULK1 and ATG16L1 over the aggresomes. ULK1 promotes autophagosome initiation, and ATG16L1 facilitates the extension/elongation of the phagophore; hence, the presence of ULK1 and ATG16L1 over the aggresomes may assist in *de novo* autophagosome biogenesis leading to the autophagic sequestration and clearance of the aggresomes. We show that TRIM16 contributes to the autophagic degradation of the protein aggregates by increasing positional availability of Ub, p62, and LC3B (the autophagosome marker protein) over the aggregates (Fig 9I). Taken together, we show that TRIM16 is a pro-survival protein, which under stress conditions utilizes all its instruments (autophagy, NRF2-p62, and Ub system) to keep the cell healthy and fit (Fig 9I).

The fact that the misfolded protein aggregates never accumulate in the healthy cell is due to the quality control system of the cell which includes ubiquitin–proteasome system (UPS), chaperone-mediated autophagy (CMA) and macroautophagy (Chen *et al*, 2011). Mainly, UPS is the first line of defense for degradation of misfolded proteins; however, the misfolded proteins which escape this quality control system can form protein aggregates that in most cases is degraded by macroautophagy process (Kirkin *et al*, 2009; Chen *et al*, 2011). The initial event which tags the misfolded proteins for subsequent degradation by UPS or CMA or autophagy is the poly-ubiquitination. The ubiquitin precursor proteins in human are encoded by *Ubb, Ubc, UBA52,* and *RPS27A* genes and *Ubb* and *Ubc,* are shown to be upregulated by

oxidative and proteotoxic stresses (Bianchi *et al*, 2015). We show that TRIM16 via NRF2 play a significant role in their upregulation. Further, a number of Ub-system genes which include E1 activation, E2 conjugation, and E3 ligation enzymes are upregulated by proteotoxic stress in TRIM16- and NRF2-dependent manner underscoring the importance of TRIM16 and NRF2 in misfolded protein ubiquitination. We show that NRF2 and its downstream genes *Ubb* and *Ube2n* are required for formation of aggresomes/ALIS and NRF2 is sufficient to complement the aggresome biogenesis defect of TRIM16-depleted cells. Although NRF2 is an immensely studied transcription factor, its function in misfolded proteins ubiquitination and biogenesis of protein aggregates was never evidently shown before this study.

Our data suggest that TRIM16 activates NRF2 in several ways. TRIM16 increases vicinity of NRF2 with p62, an event important for NRF2 activation. TRIM16 reduces NRF2 and KEAP1 interaction liberating NRF2 from the inhibitory complex. TRIM16 increases interaction between KEAP1 and p62, most likely for autophagic sequestration/degradation of KEAP1. TRIM16 decreases K48-linked ubiquitination of NRF2 (maybe by blocking KEAP1 interaction with NRF2) and increases K63-linked ubiquitination of NRF2. Both events can provide stability to NRF2 for its activation. The SPRY domain of TRIM16 is important for K63-linked ubiquitination of NRF2, ubiquitin conjugation to misfolded proteins, and protein aggregate formation suggesting that K63-linked ubiquitination is an important event associated with NRF2 stability and activation. Taken together, the data suggest that TRIM16 invokes several mechanisms to stabilize and activate NRF2.

Understanding the molecular mechanisms of protein aggregation and their clearance will provide newer potential therapeutic targets for developing efficient therapies against proteinopathies. Many neurodegenerative diseases, including Alzheimer's, Parkinson's, amyotrophic lateral sclerosis (ALS), are caused by the misfolding and aggregation of proteins such as tau, alpha-synuclein, superoxide dismutase 1 (SOD1), and TAR DNA-binding protein-43 (TDP-43; Lansbury & Lashuel, 2006). A recent study showed that TRIM16 interacts with TDP-43 (Kim *et al*, 2016), the protein which is identified as the major component of the neuronal cytoplasmic inclusion bodies deposited in ALS (Scotter *et al*, 2015). Further studies are needed to understand the physiological relevance of this interaction in ALS. Growing evidence suggests the role of protein misfolding and aggregation in cancer (Scott & Frydman, 2003; Xu *et al*, 2011). The cancer cells can maintain protein homeostasis due to intact or overactive protein aggregates turn over machinery and thus can

survive in harsh oxidative stress conditions. The TRIM16-depletion cripples these machinery leading to the accumulation of toxic misfolded proteins or aggregates intermediates and makes the cancer cell vulnerable to oxidative or proteotoxic stresses in vitro and in vivo (Fig 7). Taken together, this work highlights the significance of TRIM16 in proteostasis and hints that therapeutic modulation of TRIM16 could be a potential strategy to treat proteinopathies including cancer.

# Materials and Methods

### Cell culture

Cell lines were purchased from ATCC. HeLa (CCL-2) and human embryonic kidney (HEK293T, CRL-11268) were grown in Dulbecco's modified Eagle's medium supplemented with 10% fetal bovine serum and penicillin/ streptomycin (10,000 unit/ml). THP-1 (TIB-202) cells were grown in RPMI-1640 (Gibco) media supplemented with 10% FBS, 5 mM L-glutamine, glucose (5%), HEPES buffer, sodium pyruvate (1 mM), penicillin/streptomycin (10,000 units/ml).

### Inhibitors and concentrations

MG132 (10 μM or 20 μM, Sigma #C2211), puromycin (5 μg/ml, Sigma #P8833), $H_2O_2$ (100–400 μM, Merck), $As_2O_3$ (2.5 or 5 μM, Sigma #202673), bafilomycin A1 (300 nM, Sigma # B1793 and Invivogen # tlrl-baf1).

### Plasmids, siRNA, and transfection

GFP-polyQ74 (#40262), GFP-p62 (#38277), GFP-NRF2 (#21549), GFP-KEAP1 (#28025), Flag-KEAP1 (#28023), Myc-NRF2 (#21555), Flag-NRF2 (#36971), HA-K48 (#17605), HA-K63 (#17606), Flag-ULK1 (#27636) were procured from Addgene. GFP-TRIM16 and Flag-TRIM16 were cloned as described previously (Chauhan *et al*, 2016). Myc-TRIM16, Myc-TRIM16 deletion constructs, and His-K63-UB were generated using Gateway cloning strategy as per standard protocol (Invitrogen).

The siRNA for NRF2 and p62 were purchased from Sigma, and TRIM16, Ubb, Ube2n siRNA were from Dharmacon. For overexpression experiments, HEK293T cells were transfected using calcium phosphate method as per the manufacturer's instructions (Profection, Promega). Other cells are transfected using Effectene (Qiagen) or Viafect (Promega) or Interference (Polyplus) as per the manufacturer's instruction.

### CRISPR knock out cell line

The TRIM16 knockout cell line is generated using CRISPR-CAS9 technology and is described previously (Chauhan *et al*, 2016).

### Western blotting

The NP-40 (Invitrogen #FNN0021) or radioimmunoprecipitation assay (RIPA) buffer (20 mM Tris, pH 8.0; 1 mM, EDTA; 0.5 mM, EGTA; 0.1% sodium deoxycholate; 150 mM NaCl; 1% IGEPAL (Sigma #I8896); 10% glycerol) supplemented with protease inhibitor

cocktail (Roche) and 1 mM PMSF was used to make cell lysates. Protein concentration was measured by BCA kit (Pierce). Protein lysates were separated on SDS–polyacrylamide gel, transferred onto nitrocellulose membrane (Bio-Rad), and blocked for 1 h in 5% skimmed milk. Subsequently, membranes were incubated in primary antibody overnight at 4°C, washed (3× PBS/PBST), and then incubated for 1 h with HRP-conjugated secondary antibody (Promega). After washing with PBS/PBST (3×), the blots are developed using enhanced chemiluminescence system (Thermo Fisher). Densitometric analysis of Western blots was done using ImageJ software.

Primary antibodies used in Western blotting with dilutions were as follows: Trim16 (Santa Cruz #SC-79770; 1:1,000; Bethyl laboratories #A301-159A; 1:200), c-Myc (Santa Cruz #SC-40; 1:750), α-tubulin (Abcam #Ab7291; 1:5,000), p62 (BD #610832; 1:2,000), caspase-3 (Santa Cruz #SC-7148; 1:750), PARP1 (CST #9542L; 1:1,000 and Santa Cruz #SC-7150; 1:500), cleaved PARP1 (Santa Cruz #SC-23461; 1:500), NRF2 (Abcam #ab62352; CST-mAb #12721S, 1:2,000), GFP (Abcam #Ab290; 1:5,000), actin (Abcam #ab6276; 1:5,000). Flag (Sigma #F1804; 1:1,000), LC3B (Sigma #L7543; 1:2,000), anti-multi ubiquitin (MBL #D0583; 1:1,000) ATG-16L (Sigma #SAB140760; 1:1,000), ULK-1 (CST #8054S 1:1,000), HA (CST #3724S; 1:1,000), Keap1 (CST #8047S; 1:1,000), phospho-p62 (MBL #PM074; 1:2,000), lamin B1 (CST #12586S; 1:1,000), K48 (CST #4289S; 1:1,000), and K63 (CST #5621S; 1:1,000), His-tag (CST #2365S; 1:1,000). HRP-conjugated secondary antibodies were purchased from Santa Cruz (1:2,000) or Promega (1:5,000).

### Immunoprecipitation assay

For immunoprecipitation assays, cells were lysed in NP-40 lysis buffer (Thermo Fisher Scientific #FNN0021) supplemented with protease inhibitor/phosphatase inhibitor cocktails and 1 M PMSF for 20 min at 4°C and centrifuged. The supernatant was incubated with specific antibody at 4°C (2 h to overnight) on tube rotator followed by incubation with Protein G Dynabeads (Invitrogen, #10004D) for 2 h at 4°C. The beads were washed with ice-cold PBST (4×), and the proteins were eluted from washed beads by boiling for 5 min in 2× Laemmli sample buffer and proceeded for Western blot analysis.

### Ni-NTA pull-down assay

HEK-293T cells were transfected with desired plasmids. Next day, cells were lysed with lysis buffer containing 6 M guanidine hydrochloride for 20 min at 4°C and lysate were clarified at 12,000 *g* for 15 min. Meanwhile, 100 μl Ni-NTA slurry corresponding to 50 μl beads (Qiagen #30210) was washed with lysis buffer. Then, 700 μl of cell lysate was incubated with 50 μl of 100% Ni-NTA agarose beads for 3–4 h. The beads were sedimented at 2,000 *g*, followed by one time wash with lysis buffer and 4 times with 1× PBS. Proteins were eluted from agarose beads by boiling for 5 min in 50 μl 2× Laemmli sample buffer and immediately processed for Western blot analysis.

### Immunofluorescence and immunohistochemistry analysis

About $10^5$ cells were plated on a coverslip. The next day, cells were fixed in 4% paraformaldehyde for 10 min, permeabilized with 0.1%

saponin (or 0.1% Triton X-100) for 10 min, followed by blocking with 1% BSA for 30 min at room temperature (RT). Next, cells were incubated with primary antibody for 1 h at RT, washed thrice with 1× PBS, followed by 1 h incubation with Alexa fluor-conjugated secondary antibody. Cells were again washed thrice with 1× PBS, mounted (Prolong gold antifade, Invitrogen), air-dried, and visualized using Leica TCS SP5 confocal microscope.

For immunohistochemistry (IHC), tumor tissue samples were harvested and preserved in 10% formalin buffer solution at room temperature. Tissues were processed for paraffin embedding, and multiple 5-µm sections were prepared using microtome. Briefly, slides were deparaffinized and hydrated with deionized water. Antigen retrieval was performed in acidic pH citrate buffer (Vector Lab #H3301) by incubating the slides in a steam cooker for 20 min. Slides were kept for cooling and then washed twice with 1× PBS for 5 min, followed by endogenous peroxidase quenching in 3% hydrogen peroxide (Merck) for 15 min. Nonspecific binding was blocked by incubating the slides with horse serum (Vector Lab #PK7200) for 30 min, followed by incubation with primary antibody KI-67 (1:100) (Vector Lab #VP-RM04) at 4°C for overnight in a humidified chamber. Slides were then washed twice with PBST (0.1%) for 5 min and incubated with biotinylated anti-rabbit/mouse IgG secondary antibody (Vector Lab #PK-7200) for 45 min, followed by washing with PBST for 5 min and treated with ABC reagent (Vector Lab #PK-7200) for 30 min at room temperature. 3,3′-Diaminobenzidine (Vector Lab #SK-4100) was used to detect the immunoreactivity. Slides were subsequently counterstained with hematoxylin (Sigma #MHS1), followed by washing with Scott's tap water (Sigma #S5134). Slides were then dehydrated through sequential alcohol grading, cleared in xylene, and mounted with permanent mounting media (Vector Lab #H-5000). Stained slides were observed under a Leica DM500 light microscope, and images were taken at 40× magnifications.

Primary antibodies used in immunofluorescence experiments are ubiquitin (clone FK2 #MBL D058-3, 1:500), p62 (CST #5114, 1:100), LC3B (MBL #PM036; 1:500), Flag (Sigma #F1804, 1:500), GFP (Abcam #Ab290, 1:500); ULK1 (Santa Cruz #SC33182, 1:50), ATG16L (Abgent #AP1817, 1:25).

## Soluble and insoluble protein fractionation

Soluble-insoluble proteins were fractionated using lysis buffer containing 50 mM Tris (pH 8.0), 2% Triton X-100, 150 mM NaCl, 1 mM EDTA, 10% glycerol, protease inhibitor cocktail, and 1 mM PMSF (Fujita & Srinivasula, 2011). After centrifugation, the supernatant was used as soluble fraction and the pellet was extracted in 1% SDS to obtain insoluble protein fraction.

## Oligomerization assay

The cells were seeded in 10-cm plates and treated with MG132 (20 µM) for 2 h. Cells were lysed in PBS containing 0.5% Triton X-100, and the cell lysates were centrifuged at 6,797 *g* for 15 min at 4°C. Supernatants containing the soluble proteins were collected into new tubes (Triton X-soluble fractions). The Triton X-100-insoluble pellets were washed with PBS twice and then suspended in 200 µl PBS. The pellets were then cross-linked at room temperature for 30 min by adding 2 mM disuccinimidsuberate (DSS). The

cross-linked pellets were spun down at 6,797 *g* for 15 min and dissolved directly in non-reducing Laemmli sample buffer.

## Cycloheximide chase assay

To determine the protein stability, cycloheximide chase experiments were performed, where cells were treated with cycloheximide (100 µg/ml) and harvested at indicated time point thereafter using lysis buffer. Further, the stability of the indicated proteins was determined by Western blot analysis as described elsewhere.

## Clonogenic assay and MTT assay

For clonogenic assay, 2,000 cells were seeded in a six-well plate in triplicates. After 2 days, the cells were treated with indicated stress inducers and allowed to grow for 2–3 weeks until the colonies from single cells were formed. Cells were fixed in methanol: acetic acid (3:1) for 5 min at RT and stained with 0.5% crystal violet for 15 min and washed with water. The pictures were taken with a digital camera.

For MTT assays, 5,000 cells were seeded in triplicate in 25-cm$^2$ flask and allowed to grow for 2 days. The cells were treated with indicated stress inducers and the cells were allowed to grow for 5 more days. Subsequently, at each time point, media is removed, and cells were washed with PBS and treated with MTT (3-[4,5-dimethylthiazolyl-2-2,5-diphenyltetrazolium bromide) (5 mg/ml, Sigma) for 2 h. The dye precipitates were dissolved in DMSO, and the absorbance was measured at 570 nm.

## Annexin-V/Propidium Iodide staining

Apoptosis was measured using Annexin-V/PI double-staining method as per manufacturer's instruction (eBiosciences #88800574). Briefly, cells were treated with indicated stress inducers. At indicated time, followed by washing with 1× PBS and trypsinized to obtain a single-cell suspension, next, $10^5$ cells/ml were suspended in 100 µl binding buffer along with 5 µl Annexin-V fluorescein isothiocyanate and 5 µl propidium iodide for 30 min at room temperature, followed by acquisition on FACS Calibur (Beckton & Dickinson). The results were analyzed using Cell Quest Pro software.

## RNA isolation and quantitative real-time PCR

RNA isolation and qRT–PCR are performed exactly as described previously (Chauhan & Boyd, 2012). Briefly, total RNA was extracted using Trizol reagent according to manufacturer's protocols (Invitrogen). 2 µg of RNA was used for reverse transcription using high-capacity DNA reverse transcription kit (Applied Biosystems #4368813), and qRT–PCR was performed using Power SYBR green PCR master mix (Applied Biosystems #4367659) according to manufacturer's protocols.

## Animal experiments

The experimental protocols used in this study were approved by the Institutional Animal Ethical Review Committee. Six- to eight-week-old BALB/C-nude female mice weighing 16–20 g were maintained

under pathogen-free conditions in the animal house. The tumor xenograft model was established by subcutaneous injection of $10^7$ cells along with Matrigel (BD Corning), into right and left lower flanks. Animals were divided randomly into two groups (6 mice each) and after 15 days of injections (average tumor volume of ~100–120 mm$^3$), mice were intraperitoneally injected with PBS or As$_2$O$_3$ (5 mg/kg). Tumor volume was measured every alternative day till the mice were sacrificed. Tumors were isolated and processed for Western blotting or IHC.

**Expanded View** for this article is available online.

## Acknowledgements

This work is supported by the Wellcome Trust/Department of Biotechnology (DBT) India Alliance (IA/I/15/2/502071) fellowship, ILS core funding (Department of Biotechnology, India), and Early Career Reward (SERB, ECR/2016/000478) to Santosh Chauhan. Subhash Mehto and Swati Chauhan are supported by fellowship from SERB (NPDF, PDF/2016/001697) and DST (SR/WOS-A/LS-9/2016). We gratefully acknowledge the guidance, reagents, and support provided by Dr. Vojo Deretic (University of New Mexico, USA), Dr. Terje Johansen (University of Tromso, Norway), and Dr Rupesh Das (ILS, India) during the course of this study. We acknowledge the technical assistance of Mr. Kshitish Rout and Mr. Bhabani Sahoo (Microscopy facility). We gratefully acknowledge the support of ILS central facilities (Animal, Microscopy, FACS, and Sequencing).

## Author contributions

Santosh C and KKJ designed the experiments. KKJ, SPK, SM, PN, BD, PKS, AA, Swati C, and Santosh C performed the experiments. Santosh C, Swati C, and SS supervised experimental work. Santosh C, Swati C, GHS, and SKR wrote the article. All authors contributed to data analysis and commented on the manuscript.

## Conflict of interest

The authors declare that they have no conflict of interest.

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
