## [Review Process File · The EMBO Journal]

TRIM16 controls assembly and degradation of protein aggregates by modulating the p62-NRF2 axis and autophagy

Kautilya Kumar Jena, Srinivasa Prasad Kolapalli, Subhash Mehto, Parej Nath, Biswajit Das, Pradyumna Kumar Sahoo, Abdul Ahad, Gulam Hussain Syed, Sunil K Raghav, Shantibhusan Senapati, Swati Chauhan and Santosh Chauhan.

Review timeline:

Submission date:	4 th October 2017
Editorial Decision:	8 th November 2017
Revision received:	18 th March 2018
Editorial Decision:	17 th April 2018
Revision received:	25 th June 2018
Accepted:	26 th July 2018

Editor:

Transaction Report:

1st Editorial Decision

8th November 2017

Thank you for submitting your manuscript on TRIM16 in aggresome turnover for our editorial consideration. We have now received the reports from three expert referees, copied below for your information. Given that all referees consider your findings of potential interest and importance, we would in principle be interested to consider a revised version of this manuscript further for The EMBO Journal. Nevertheless, in light of the significant number of major concerns, it is also clear that various aspects of the study would need to be substantially revised before publication may be warranted. One particularly important issue would be to further explore/clarify the molecular functions of TRIM16, including possible scaffolding roles as well as the possibility that TRIM16 may itself undergo degradation during the process. The referees furthermore emphasize the importance of reorganizing and streamlining the manuscript to make it easier to understand also for non-specialist readers.

Since it is our policy to allow only a single round of major revision, I would in this case like to invite you to contact me with a tentative point-by-point response and an outline on how you intend to address the key concerns raised by all three reviewers already during the early stages of your revision work, so that I would be able to provide some feedback and guidance. Furthermore, we might discuss possible extension of the revision period (beyond the regular three months), during which time the publication of any competing work elsewhere would have no negative impact on our final assessment of your own study.

REFeree REPORTS.

Referee #1:

The finding that TRIM16 regulates the p62-Keap1-Nrf2 axis is of potential interest, and the quality

of data presented in each Figure is not so poor. However, a significant problem is that the authors presented multi and complicated functions of TRIM16 without showing consistency. This reviewer thinks that general readers cannot understand this study correctly. In addition, due to a large number of data and lack of unity of the models in each Figure, general readers will get bored halfway through reading this important work, at least current version. I recommend that the authors reconsider structure of this manuscript and focus on main point (e.g., regulation of the p62-Keap1-Nrf2 axis by TRIM16).

Specific comments

1. According to the author's claim, increased level of TRIM16 results in formation of p62- and UB-positive aggregates, Nrf2-activation, and induction of autophagy. Does simple overexpression of TRIM16 mimic this?
2. In Figure 2W and Figure 3S: the authors claimed that TRIM16 interacts with both p62 and Nrf2 to stabilize and activate Nrf2. Is Nrf2 in complex with p62 and TRIM16 translocated into nucleus? Alternatively, is Nrf2 released from the complex consisting of p62, TRIM16 and Nrf2 and then translocated into nucleus?
3. The authors should examine the level of nuclear Nrf2 in TRIM16-knockout cells.
4. p62 is one of autophagy-substrate, and Keap1 is also mainly degraded by autophagy in a p62-dependent manner (Komatsu et al., Cell 2007 and Taguchi et al., PNAS 2012), which are inconsistent with the presented data.
5. Phosphorylation of p62 at S351 (in the case of human, S349) is indispensable for the p62-mediated Nrf2-activation (Ichimura et al., Mol. Cell 2013). The authors should investigate the phosphorylation status in their experimental settings.

Referee #2:

In their manuscript, Jena et al describe a study on a member of the tripartite motif family (TRIM) proteins, TRIM16, which has recently been implicated in the process of autophagy (Chauhan, Kumar et al., 2016). Building on the previous findings that TRIM16 is able to organize protein networks, i.e. TRIM16 binds Galectin-3, TFEB, ULK1, Beclin-1, and ATG16L1 involved in autophagy (Chauhan et al., 2016), the authors now demonstrate that TRIM16 is essential for maintenance of protein aggregates in cells undergoing proteotoxic and oxidative stress. Using cells deficient in or overexpressing TRIM16, Jena et al show that TRIM16 affects ubiquitylation and stability of the transcription factor NRF2 responsible for the expression of anti-oxidant genes and those encoding proteins regulating the ubiquitylation pathway. According to this study, TRIM16 also seems to affect oligomerization and function of the selective autophagy receptor p62/SQSTM1, which plays a role both in NRF2 activation and protein aggregation and which itself is encoded by an NRF2-responsive gene. Interestingly, overexpression of NRF2 rescues protein aggregation in cells that are deficient in TRIM16 providing indirect support to the authors' claim that TRIM16 regulates aggregate formation via the NRF2 signaling axis. Jena et al further claim that TRIM16 not only promotes aggregate formation upon proteotoxic and oxidative stress but also fosters their degradation by selective autophagy. It however remains unclear how TRIM16 stimulates autophagy in this context, as evidence for recruitment of core autophagic machinery (ULK1, Beclin-1, and ATG16L1) to protein aggregates specifically via TRIM16 is limited. Finally, the authors provide some evidence that TRIM16 is essential for cell survival during oxidative and proteotoxic stress, with TRIM16 knockout (KO) HeLa cells demonstrating fragmented and depolarized mitochondria and higher levels of apoptosis. This decreased fitness of TRIM16 KO cells translates in their decreased survival as tumor xenografts in nude mice. The authors thus conclude that by organizing protein aggregates and by propagating the NRF2 signaling, TRIM16 is an essential factor in the cellular defense system against the proteotoxic and oxidative stress.

The study presented by Jena et al is complex and attempts to draw strong conclusions from (still) limited experimental evidence (plentiful experiments do not always go deep enough into the questions they address). Thus, additional evidence is required to understand how the atypical TRIM protein (e.g., TRIM16 lacks the RING domain which is usually required for the E3 ubiquitylation activity) can play such versatile functions - organize protein aggregates, mediate protein ubiquitylation and convey signals to the nucleus (gene transcription effects). The role of individual domains of TRIM16 in these functions requires additional analysis to exclude artefacts and provide

biochemical rationale for the observed biological effects. Finally, more controls are required to prove specificity of the claimed interactions for the TRIM16. Once these concerns have been addressed, the very interesting biology of TRIM16 can be better understood at the molecular level. I therefore recommend thorough revision of the manuscript before it can be considered for publication. Below are specific points that need to be addressed prior to resubmission.

Specific points:

1. TRIM16 is required for the biogenesis of protein aggregates

- The authors convincingly demonstrate that TRIM16 KO HeLa cells do not support formation of Ubiquitin (Ub)- and p62-positive protein aggregates upon treatment with H₂O₂ (oxidative stress) and MG132 or puromycin (proteotoxic stress). A plethora of assays (imaging and cell fractionation) were performed to that effect (Fig. 1). It is however unclear whether TRIM16 itself acts as a scaffold in protein aggregation. Does endogenous TRIM16 localize with Ub, p62 or LC3B in the protein aggregates? Is TRIM16 recovered in the detergent-insoluble fraction? These are important questions, as the authors use TRIM16 reconstitution/overexpression throughout their study but do not comment on whether TRIM16 is itself an aggregation-prone protein or one whose role may be to scaffold protein aggregates. A scaffolding role could clearly provide an explanation for the involvement of TRIM16 in protein aggregation.

2. TRIM16 interacts with p62 and NRF2 via its SPRY domain

- TRIM16 is known to interact with a plethora of proteins (Chauhan et al., 2016, Kimura, Jia et al., 2017). In the current study, the authors show that TRIM16 binds p62, NRF2 and KEAP1. Using co-IP, the authors map the interactions to the C-terminal SPRY domain of TRIM16 (Fig. 2P-U). Is this interaction direct? It is recommended to use purified TRIM16 (or its SPRY domain) and purified p62 or NRF2 to prove the direct binding. Which domains of p62 and/or NRF2 would interact with TRIM16? Would TRIM16 compete for NRF2 or p62 binding with KEAP1?
- It would be important to include a negative control in the direct protein binding assay (e.g., GST alone in the GST pulldown assay) to demonstrate specificity of the TRIM16 interactions. Does the SPRY domain interact with itself or is that the coiled-coil domain (CCD) only that would support potential TRIM16 oligomerization?

3. TRIM16 fosters p62 oligomerization

- The authors suggest that TRIM16 affects protein aggregate formation in part by fostering p62 oligomerization. Without looking into the possible scaffolding role of TRIM16, they show that TRIM16 reduces the K63-ubiquitylation status of p62 which might lead to increased p62 oligomerization. The authors refer to the study by Pan et al, in which another TRIM family member, TRIM21, promoted K63-linked ubiquitylation within the PB1 domain of p62, which is responsible for p62 oligomerization (Pan, Sun et al., 2016). However, in their own study, the authors do not provide strong evidence for the changes in p62 oligomerization. The Flag/GFP-p62 interaction assay (Fig. 3F), although indicative of the effect, does not provide reliable information. Why would GFP-p62 and Flag-p62 interact in the cell poorly in the first place? The PB1 domain of p62 should ensure intermolecular interaction between p62 molecules attached to different tags. Endogenous NBR1 is the protein that might interfere with the cellular p62 oligomerization assays. NBR1 possesses a PB1 domain that interacts with the PB1 domain in p62 (Kirkin, Lamark et al., 2009). In the least, the potential effect of NBR1 needs to be excluded from the system (e.g., by using p62 mutants that do not bind NBR1 or using NBR1-depleted cells). Biochemical assays using purified proteins might be a resource-intensive alternative to the cellular assays.
- If TRIM16 binds p62 directly, does this per se promote p62 oligomerization (irrespective of p62 ubiquitylation status)? Results obtained with overexpression of p62 in TRIM16 KO HeLa cells (Fig. 5A-D) do not support this notion so far and are at odds with the authors' preceding finding (Fig. 3F). An experiment using purified TRIM16 and p62 could cast light on this issue.

4. TRIM16 activates NRF2 signaling

- The authors claim that TRIM16 is found in a complex with p62, KEAP1 and NRF2 (Fig. 2). Of interest, while the TRIM16-KEAP1 interaction was not affected (Fig. S2G), the TRIM16-p62 complex (Fig. 2O) was strengthened and the TRIM16-NRF2 complex was weakened (Fig. 2S vs. Fig. S2D) after the MG132 treatment. As shown previously, p62 competes with NRF2 for binding to KEAP1, liberating NRF2 from its inhibitory/degradative complex with KEAP1 and thereby activating the NRF2 signaling (Jain, Lamark et al., 2010, Komatsu, Kurokawa et al., 2010). So, how do the authors envisage the way TRIM16 would fit into this well-known scheme? Does TRIM16

oust KEAP1 from its complex with NRF2, as is the case for p62? Or, is TRIM16's role merely to bring more p62 into the close proximity with NRF2 (the scaffolding role)?

- TRIM16 seems to affect NRF2 ubiquitylation status (Fig. 3A-B). Since the TRIM16-KEAP1 complex is not affected by the MG132 treatment, and KEAP1 is the major E3 ligase for NRF2, it is presently not clear what causes the change in NRF2 ubiquitylation pattern. To claim a TRIM16-mediated activating effect on NRF2, the authors must provide a mechanistic explanation using their own data. Current explanations are speculative and insufficient.

5. TRIM16 drives transcription of NRF2-responsive genes

- The authors show that the transcription of a number of NRF2-responsive genes (upon oxidative/proteotoxic stress) is significantly reduced in TRIM16 KO cells (e.g., Fig. 3L-N). This is suggested to be due to the scaffolding role of TRIM16 in stabilizing NRF2, p62 and protein aggregates. It would therefore be important to test if TRIM16 overexpression alone (in the absence of the proteotoxic or oxidative stress!) is sufficient to induce NRF2 gene response. Presumably, this should not be the case in NRF2 KO cells.

6. TRIM16 stimulates autophagy

- The authors propose that TRIM16 positively regulates autophagy. This seems to be in line with the previous report (Chauhan et al., 2016). However, it is unclear whether TRIM16 is itself degraded by autophagy. Use of the well-characterized double tag (RFP-GFP) attached to TRIM16 to monitor its lysosomal distribution could help answer this question.
- If TRIM16 is overexpressed, it does occasionally form aggregates (e.g., Fig. 5K) suggesting its propensity to self-associate. Do these aggregates become larger if autophagy is inhibited (e.g., in bafilomycin A1-treated cells)? What is the role of the p62/NBR1 system in TRIM16 aggregation. Do TRIM16 aggregates become larger or smaller if p62 and/or NBR1 are absent from the system?
- The data on mitophagy are preliminary and can be removed from the manuscript, unless proper dissection of the role of TRIM16 in mitochondrial ubiquitylation or on expression of mitophagy receptors is performed.

7. TRIM16 affects cell survival during oxidative and proteotoxic stress

- The use of AS2O3 in cell and xenograft assays, comes a bit as a surprise at the end of the manuscript. All mechanistic studies were performed using H₂O₂, MG132 and Puromycin. Why not to use the clinically relevant proteasomal inhibitor Velcade/Bortezomib in the cell and xenograft studies to complete the experiments in one line?

8. Structure of the manuscript

- The Introduction and Discussion sections can be more detailed, while some passages from the Results section need to be moved in the Introduction. Similarly, some speculative passages in the Results section should be moved into the Discussion section.

References:

- Chauhan S, Kumar S, Jain A, Ponpuak M, Mudd MH, Kimura T, Choi SW, Peters R, Mandell M, Bruun JA, Johansen T, Deretic V (2016) TRIMs and Galectins Globally Cooperate and TRIM16 and Galectin-3 Co-direct Autophagy in Endomembrane Damage Homeostasis. *Dev Cell* 39: 13-27
- Jain A, Lamark T, Sjøttem E, Larsen KB, Awuh JA, Overvatn A, McMahon M, Hayes JD, Johansen T (2010) p62/SQSTM1 is a target gene for transcription factor NRF2 and creates a positive feedback loop by inducing antioxidant response element-driven gene transcription. *J Biol Chem* 285: 22576-91
- Kimura T, Jia J, Kumar S, Choi SW, Gu Y, Mudd M, Dupont N, Jiang S, Peters R, Farzam F, Jain A, Lidke KA, Adams CM, Johansen T, Deretic V (2017) Dedicated SNAREs and specialized TRIM cargo receptors mediate secretory autophagy. *EMBO J* 36: 42-60
- Kirkin V, Lamark T, Sou YS, Bjorkoy G, Nunn JL, Bruun JA, Shvets E, McEwan DG, Clausen TH, Wild P, Bilusic I, Theurillat JP, Overvatn A, Ishii T, Elazar Z, Komatsu M, Dikic I, Johansen T (2009) A role for NBR1 in autophagosomal degradation of ubiquitinated substrates. *Mol Cell* 33: 505-16
- Komatsu M, Kurokawa H, Waguri S, Taguchi K, Kobayashi A, Ichimura Y, Sou YS, Ueno I, Sakamoto A, Tong KI, Kim M, Nishito Y, Iemura S, Natsume T, Ueno T, Kominami E, Motohashi H, Tanaka K, Yamamoto M (2010) The selective autophagy substrate p62 activates the stress responsive transcription factor Nrf2 through inactivation of Keap1. *Nat Cell Biol* 12: 213-23

Pan JA, Sun Y, Jiang YP, Bott AJ, Jaber N, Dou Z, Yang B, Chen JS, Catanzaro JM, Du C, Ding WX, Diaz-Meco MT, Moscat J, Ozato K, Lin RZ, Zong WX (2016) TRIM21 Ubiquitylates SQSTM1/p62 and Suppresses Protein Sequestration to Regulate Redox Homeostasis. *Mol Cell* 61: 720-733

Referee #3:

In this study, Jena and co-workers show that TRIM16 plays a dual role in the cellular response to aggregated proteins by stimulating their sequestration in inclusion bodies as well as the clearance of aggregated proteins by macroautophagy. These are new roles for TRIM16 and the authors argue that the p62-NRF2 axis revealed in this study links the anti-oxidative response to ubiquitin-dependent clearance of misfolded proteins. The authors provide an extensive amount of data supporting this model using mostly microscopic and biochemical tools with which they study this process in cell-based systems. Moreover, they also the importance of this response for tumor growth in a xenotransplant mouse model. The finding is novel and the study is detailed and overall convincing.

Major concerns

Title and conclusions of the paper. Can the authors exclude the possibility that TRIM16 does not induce degradation of aggresomes by autophagy but the formation of aggresomes by facilitating degradation of protein aggregates or aggregation-prone proteins that would otherwise end up in the aggresome. Note that aggresomes and protein aggregates are not the same thing. If this possibility cannot be conclude the title should be phrased more carefully as it suggests now that the aggresomes themselves are turned over.

Fig. 1J Does TRIM16 knockout affect the levels of total ubiquitin conjugates? If so, that could be the reason for a reduction in ubiquitin-positive inclusions in TRIM16 knockout cells. Is there a selective effect on total/soluble/insoluble levels of K48 or K63 chains?

Fig S1M-O. I don't understand the rational for the choice of ULK1 and Beclin1 as representative substrates for monitoring inclusions. Both proteins are also involved in macroautophagy. Why did the authors select these proteins? It would be better to use an aggregation-prone substrate that is not involved in clearance of misfolded proteins as a reference.

Fig 1P. if I am not mistaken, proteostat has an excitation wavelength of around 480 nm. The authors did not mention the Alexa dye that was used in that experiment but I suspect that p62 was stained with an Alexa488 dye (because it is depicted green in the micrograph). If so, there is a serious risk for spectral contamination when combining these dyes because they will be excited with the same wavelength. Careful selection of emission filters may not entirely solve this problem. Therefore, it would be helpful if the authors (for example in the supplementary figures) show micrographs for proteostat and p62 of cells that have only been stained with proteostat or for p62.

Fig 2M-N. The destabilization of p62 is not very convincing. On how many experiments are the curves shown in Fig 2N based? Include error bars.

Fig 2O. Not only the input but also pulldown of TRIM16 should be shown for the immunoprecipitations. It is not clear for me why the blot is cut after the lanes that show the input for TRIM16.

Fig 3A, B, D, F. There are several controls missing in these pulldown experiments. First, these experiments should include pulldowns from lysates of cells that express plain GFP (for GFP-NRF2) or mock transfected cells (for epitope tagged p62) to determine the specificity of the pulldown. Second, the western blot should not only been shown for the input of the precipitated protein (GFP-NRF2 or epitope tagged p62) but also for the pulldown. Third, molecular weight markers should be depicted. This is of particular interest for Fig 3A and B as it would allow evaluating if the ubiquitin smear localizes at the correct molecular weight for ubiquitylated GFP-NRF2.

Fig 3D. p62 is a ubiquitin-binding protein and there is a risk that the signal on the ubiquitin blot comes from ubiquitylated proteins that were co-immunoprecipitated with FLAG-p62 and not from

ubiquitylated p62. Was the sample denatured before the immunoprecipitation was performed? Can the authors exclude that the signal comes from ubiquitylated proteins bound to p62?

Fig 3Q. It appears that p62 immunostaining signal is also dramatically decreased when UBB or Ube2N are depleted. Is there a reduction in the total levels of p62 when these genes are knocked down? This can be checked by western blotting.

Fig 4I. I guess that the difference in the molecular weight of endogenous NRF2 and myc-NRF2 is too little to detect this as independent bands in the Nrf2 blot. Even if this is the case, I would still expect an increase in the NRF2 band intensity because of the overexpression of myc-NRF2. Please explain why this is not observed. It would also be helpful to show a larger part of the NRF2 blot so that the GFP-NRF2 can be seen in the same NRF2-probed blot. Also here it would be helpful if molecular weight markers are included.

Fig 5N. The patterns for ULK1 and ATG16L1 are very different. Isn't that surprising since they are both acting on early autophagosomes and are therefore expected to co-localize?

TRIM16 associates with proteins like p62 that localize in the autophagosome and are degraded in the process of targeting substrates for degradation. Is TRIM16 degraded by autophagy?

Minor concerns

Abstract. The opening sentence is too generic. The authors talk about "a protective effect of assembly of aggregates from misfolded proteins". While it is rather well established that the sequestration of protein aggregates in inclusion bodies/aggregates has a beneficial effect, the actual aggregation of misfolded proteins may well be linked to their toxicity and not protective.

"TRIM16 interacts, stabilizes, affects K63-linked ubiquitination, and increases the oligomerization capacity of p62". Grammar incorrect, sentence unclear.

Introduction

"Normally, these misfolded proteins are ubiquitinated and are degraded by the proteasome system or chaperone-mediated autophagy (Kraft et al., 2010)." What about macroautophagy? The review that is cited discusses primarily clearance of misfolded proteins by ubiquitin-selective (macro)autophagy. It is better to cite the original papers but if necessary and reviews need to be cited they have to be correct. Here references/reviews on the role of CMA and the proteasome in clearance of misfolded should be included.

A few recent papers on TRIM16 and (secretory) autophagy that appear to be relevant for this study are not cited/discussed. In particular, Fraiberg and Elazar, 2016, *Developmental Cell*; Kimura et al, *Autophagy*, 2017; Kumar et al, *Autophagy*, 2017 and Kimura et al, *EMBO Journal*, 2017.

Results

"dot's" should be dots.

Discussion

"autophagasomes" should be autophagosomes.

"The poly-ubiquitin proteins in human are encoded by Ubb and Ubc genes". I would rather talk about ubiquitin precursor proteins so that it is not confused polyubiquitin conjugates. Note that there are two more ubiquitin precursor proteins in the human genome.

Please check for grammar and spelling.

Referee#1:

The finding that TRIM16 regulates the p62-Keap1-Nrf2 axis is of potential interest, and the quality of data presented in each Figure is not so poor. However, a significant problem is that the authors presented multi and complicated functions of TRIM16 without showing consistency. This reviewer thinks that general readers cannot understand this study correctly. In addition, due to a large number of data and lack of unity of the models in each Figure, general readers will get bored halfway through reading this important work, at least current version. I recommend that the authors reconsider structure of this manuscript and focus on main point (e.g., regulation of the p62-Keap1-Nrf2 axis by TRIM16).

We are very thankful to the reviewer for the careful assessment of the manuscript. We understand the reviewer concern regarding “consistency” and “unity” due to extensive data presented in this manuscript. We have now reduced the amount of data and have focused on the main points as suggested.

The work presented in the manuscript shows the importance of TRIM16 in both biogenesis and degradation of protein aggregates. We have no problem in showing only the data pertaining to the role of TRIM16 in regulation of p62-Keap1-NRF2 axis and biogenesis of protein aggregates. However, in that case, several of the comments of reviewer 2 and 3 would have become purposeless and probably the other reviewers may not like this. Besides, we ourselves think that if we would have remove the data regarding the role of TRIM16 in protein aggregates degradation, we may not be able to justify the phenomenal biological function of TRIM16 in protein aggregates turnover. So, we have taken a midway, we have restructured the manuscript to make it more streamlined. After thorough discussion with editor, we have removed several parts of manuscript which are not going deep enough and also we have strengthened the other parts which are main theme of manuscript (esp. scaffolding role of TRIM16 and regulation of p62-Keap1-NRF2 axis by TRIM16). Now, we can explain our manuscript in one line "TRIM16 regulates protein aggregates biogenesis and degradation by modulating p62-Keap1-NRF2 axis and autophagy"

Following changes are made:

(1) We have removed data which appears to be reducing the unity of the manuscript:

- We have removed the data pertaining to the role of TRIM16 in mitochondrial health (Figure 6A and 6B, Supplementary Figure 6A in old manuscript version).*
- We have removed the data regarding the cross-talk between TRIM21 and TRIM16 and oligomerization of p62 (Figure 3D-3F in old manuscript version).*
- We have removed the data regarding biogenesis of aggregates of overexpressed Beclin1 and ULK1 (Supplementary Figure 1M-O).*

(2) Several new experiments are performed to strengthen the main theme of the manuscript:

- *Scaffold role of TRIM16: Working on comments of Reviewer 2 and 3, several new experiments are performed to show that TRIM16 act as scaffolding protein. New experiments are performed to strengthen the data of TRIM16 interaction with aggregation and autophagy machinery proteins (interaction with p62, LC3B and Ubiquitin).*
- *Regulation of NRF2-KEAP1-p62: Several new experiments are performed in the revised manuscript to further increase our understanding of TRIM16-mediated regulation NRF2-KEAP1-p62.*

Specific comments

1. According to the author's claim, increased level of TRIM16 results in formation of p62- and UB-positive aggregates, Nrf2-activation, and induction of autophagy. Does simple overexpression of TRIM16 mimic this?

Answer: Reviewer concerns are divided in parts so that we can answer it clearly. Please see answer to each concern below.

Q. Does the overexpression of TRIM16 results in formation of p62- and UB-positive aggregates?

Ans. The figure 7I (also showed in right panel here) shows that overexpression of TRIM16 does results in formation of p62- and UB-positive aggregates but the number of cells having these aggregates were less and the aggregates were small (Figure 7L). This insufficiency could be because of two reasons (1) just overexpression of TRIM16 in normal cells (un-stressed) may not trigger protein misfolding and protein aggregates formation. (2) the equilibrium between formation and degradation of these aggregates might have shifted toward degradation due to overexpression of TRIM16 which can induce autophagy. So, we performed the experiments in presence of MG132 (to induce protein aggregates formation) and BafilomycinA1 (to reduce autophagy mediated degradation of aggregates) (Figure 7I and 7J). In these conditions, the TRIM16 overexpressing cells showed significant number of huge p62- and UB-positive aggregates which co-localizes with GFP-TRIM16 (Figure 7I and 7J). This work is detailed in new manuscript at page no. 12 and paragraph 2.

Q. Does the overexpression of TRIM16 results in NRF2-activation?

Ans. Yes. We overexpressed TRIM16 and performed the qRT-PCR to determine the status of NRF2-activated genes (**New Experiment no. 1**). The overexpression of TRIM16 increased the expression of the key NRF2-regulated genes, *Ho-1*, *p62*, and *Nqo1*. This increased expression was blunted in absence of NRF2 (siRNA knock down of NRF2) suggesting that this enhanced expression is mediated via NRF2. These experiment are presented in (Figure 5 H-J) and below.

Q. Does the overexpression of TRIM16 results in autophagy induction?

Ans. The data in Expanded View Figure 1L, 1M and 2E (old manuscript, Supplementary Figure 1K and Supplementary Figure 2B) show that overexpression of TRIM16 increases the LC3B amount (the autophagy marker protein). Further, we also show that cells depleted of TRIM16 are attenuated for increase in autophagy flux (Figure 1N and 1O).

2a. In Figure 2W and Figure 3S: the authors claimed that TRIM16 interacts with both p62 and Nrf2 to stabilize and activate Nrf2. Is Nrf2 in complex with p62 and TRIM16 translocated into nucleus?

Ans. In our assays (stressed or unstressed), we never found that TRIM16 or p62 translocates to the nucleus (Figure. 7I and Expanded View Figure 2B and also see the new Experiment no. 2 below). TRIM16 appears to be like p62, which is known to stabilize NRF2 but does not translocates to the nucleus with NRF2.

2b. Alternatively, is Nrf2 released from the complex consisting of p62, TRIM16 and Nrf2 and then translocated into nucleus?

Ans. Yes, reviewer is absolutely correct.

3. The authors should examine the level of nuclear Nrf2 in TRIM16-knockout cells.

Ans. We examined the levels of nuclear NRF2 in TRIM16^{KO} cells compared to the control cells. The total amount of nuclear NRF2 was lesser in TRIM16^{KO} cells (**New Experiment no. 2, Right panel**). The experiment is presented in Figure EV2B.

4. p62 is one of autophagy-substrate, and Keap1 is also mainly degraded by autophagy in a p62-dependent manner (Komatsu et al., Cell 2007 and Taguchi et al., PNAS 2012), which are inconsistent with the presented data.

Ans. Please pardon us here but probably we are not able to understand that how our study is inconsistent with the indicated studies. We think that our data is very much consistent with these previous studies. Komatsu et al 2007 showed that “Depletion of p62 suppresses the appearance of ubiquitin-positive protein aggregates in cells, indicating that p62 plays an important role in inclusion body formation”. Our data also showed that TRIM16 is required for stability and the expression of p62 and in the absence of TRIM16, protein aggregates are not formed properly (one of the reason is low p62/NRF2 in these cells). The overexpressed p62 protein which is known to form aggregates (self-aggregating) and these aggregates are target of autophagy. We showed that these p62 aggregates are accumulated in TRIM16 knock out cells (autophagy-deficient cells) more than control cells indicating that p62 aggregates are targeted by autophagy (in agreement with Komatsu et al., Cell 2007). Our data also show that TRIM16 destabilizes KEAP1 by increasing p62 stability and also by increasing autophagy (in agreement with Taguchi et al., PNAS 2012). So, we think we are in full agreement with these publications.

5. Phosphorylation of p62 at S351 (in the case of human, S349) is indispensable for the p62-mediated Nrf2-activation (Ichimura et al., Mol. Cell 2013). The authors should investigate the phosphorylation status in their experimental settings.

Ans. The phosphorylation of p62 (S349) is increased on treatment of the cells with MG132 (New Experiment no. 3, left panel). This phenomenon was not observed in TRIM16^{KO} cells indicating that TRIM16 is important for phosphorylation of p62. This results is presented in Figure 2C and

discussed at page number 6 and paragraph 2.

Referee #2:

In their manuscript, Jena et al describe a study on a member of the tripartite motif family (TRIM) proteins, TRIM16, which has recently been implicated in the process of autophagy (Chauhan, Kumar et al., 2016). Building on the previous findings that TRIM16 is able to organize protein networks, i.e. TRIM16 binds Galectin-3, TFEB, ULK1, Beclin-1, and ATG16L1 involved in autophagy (Chauhan et al., 2016), the authors now demonstrate that TRIM16 is essential for maintenance of protein aggregates in cells undergoing proteotoxic and oxidative stress. Using cells deficient in or overexpressing TRIM16, Jena et al show that TRIM16 affects ubiquitylation

and stability of the transcription factor NRF2 responsible for the expression of anti-oxidant genes and those encoding proteins regulating the ubiquitylation pathway. According to this study, TRIM16 also seems to affect oligomerization and function of the selective autophagy receptor p62/SQSTM1, which plays a role both in NRF2 activation and protein aggregation and which itself is encoded by an NRF2-responsive gene. Interestingly, overexpression of NRF2 rescues protein aggregation in cells that are deficient in TRIM16 providing indirect support to the authors' claim that TRIM16 regulates aggregate formation via the NRF2 signaling axis. Jena et al further claim that TRIM16 not only promotes aggregate formation upon proteotoxic and oxidative stress but also fosters their degradation by selective autophagy. It however remains unclear how TRIM16 stimulates autophagy in this context, as evidence for recruitment of core autophagic machinery (ULK1, Beclin-1, and ATG16L1) to protein aggregates specifically via TRIM16 is limited. Finally, the authors provide some evidence that TRIM16 is essential for cell survival during oxidative and proteotoxic stress, with TRIM16 knockout (KO) HeLa cells demonstrating fragmented and depolarized mitochondria and higher levels of apoptosis. This decreased fitness of TRIM16 KO cells translates in their decreased survival as tumor xenografts in nude mice. The authors thus conclude that by organizing protein aggregates and by propagating the NRF2 signaling, TRIM16 is an essential factor in the cellular defense system against the proteotoxic and oxidative stress.

The study presented by Jena et al is complex and attempts to draw strong conclusions from (still) limited experimental evidence (plentiful experiments do not always go deep enough into the questions they address). Thus, additional evidence is required to understand how the atypical TRIM protein (e.g., TRIM16 lacks the RING domain which is usually required for the E3 ubiquitylation activity) can play such versatile functions - organize protein aggregates, mediate protein ubiquitylation and convey signals to the nucleus (gene transcription effects). The role of individual domains of TRIM16 in these functions requires additional analysis to exclude artefacts and provide biochemical rationale for the observed biological effects. Finally, more controls are required to prove specificity of the claimed interactions for the TRIM16. Once these concerns have been addressed, the very interesting biology of TRIM16 can be better understood at the molecular level. I therefore recommend thorough revision of the manuscript before it can be considered for publication. Below are specific points that need to be addressed prior to resubmission.

We are very thankful to the reviewer for studying the manuscript thoroughly and for extensive constructive comments. These comments have helped us to make this manuscript much stronger and also have opened new avenues for further analysis of the topic in new manuscripts. In this manuscript, now we present an extensive amount of data showing that TRIM16 act as a scaffold protein and is important for protein aggregates biogenesis and degradation. At a molecular level, we show that how TRIM16 modulate NRF2-Keap1-p62 axis. This is a very detailed analysis, where we showed that TRIM16 interacts with NRF2, Keap1 and p62, affects their

ubiquitination, regulates their stability and modulates their interactions. Further, we found that TRIM16 has tendency to oligomerize/aggregate and participates in protein aggregates formation by modulating p62 expression and affecting the transcription of ubiquitin genes (via NRF2). Domain deletion experiments were performed to identify (1) the domain of TRIM16 required for interaction with NRF2 and p62, (2) the domain of TRIM16 required for its oligomerization, (3) the domain of TRIM16 required for K63-linked ubiquitination of NRF2, (4) the domain of TRIM16 required for protein aggregation. Our data also showed that NRF2 is the prime mediator for TRIM16 mediated protein aggregates biogenesis response. The knocking down NRF2 or its targeted genes affects protein aggregates formation capacity. We have also deciphered a positive feedback loop TRIM16-NRF-p62. We further showed that TRIM16 is important for mediating autophagic degradation of protein aggregates by organizing assembly of autophagy proteins over the protein aggregates. We didn't stop here and showed the physiological relevance of current finding both in-vitro (cell death assays) and in-vivo (tumorigenesis).

The reviewer 1 and 3 thinks that work presented (in our previous manuscript) is quite extensive. In this revision process, we have performed 22 new experiments (now manuscript has 9 main Figures, 5 Expanded View Figures and 3 Appendix Supplementary Figures). The "reviewer 1" wants us to reduce the data for the sake of clarity and unity. The reviewer 2 also pointed out that some of the results does not go deep enough. Being a new investigator, I may have got over excited and presented data which is preliminary (this was my fault). Hence, after thorough discussion with editor, few of the data sets which are not going deep enough (or confusing) are removed (discussed below in specific comments). We feel that it is impossible for us to define the complete biology of a multifunctional protein like TRIM16 in a single manuscript. For example, it took more than 10 good publications to describe how p62 functions in protein aggregates biogenesis, degradation, and tumorigenesis and still there are several unanswered questions in p62 biology. This is the first manuscript of TRIM16 in terms of its role in protein aggregate biology. To answer most of the concerns of this reviewer, we have now performed 12 new experiments (~20 panels).

Specific points:

1. TRIM16 is required for the biogenesis of protein aggregates

The authors convincingly demonstrate that TRIM16 KO HeLa cells do not support formation of Ubiquitin (Ub)- and p62-positive protein aggregates upon treatment with H₂O₂ (oxidative stress) and MG132 or puromycin (proteotoxic stress). A plethora of assays (imaging and cell fractionation) were performed to that effect (Fig. 1).

Ans: Thanks for the reviewer's comment.

It is however unclear whether TRIM16 itself acts as a scaffold in protein aggregation. Does endogenous TRIM16 localize with Ub, p62 or LC3B in the protein aggregates? Is TRIM16 recovered in the detergent-insoluble fraction? These are important questions, as the authors use TRIM16 reconstitution/overexpression throughout their study but do not comment on

whether TRIM16 is itself an aggregation-prone protein or one whose role may be to scaffold protein aggregates. A scaffolding role could clearly provide an explanation for the involvement of TRIM16 in protein aggregation.

Ans. We agree with the reviewer. In this revised manuscript, we have performed several new experiments suggesting that TRIM16 has scaffolding properties and can oligomerize to form aggregates along with p62. This scaffolding role of TRIM16 is now discussed in revised version of manuscript.

For clarity reason, Reviewer's above set of questions are answered in parts:

Reviewer Question: Does endogenous TRIM16 localize with Ub, p62 or LC3B in the protein aggregates?

*Ans. The overexpressed TRIM16 nicely co-localizes with endogenous Ub (Figure 7I and 7J), endogenous p62 (Figure 7I and 7J) and endogenous LC3B (Figure 7K, **New experiment 4, see below panel on left**) at protein aggregates. We have tried several commercially available TRIM16 antibodies (catalogue number TRIM16 (sc-79770 from Santa Cruz Biotechnology; A301-160A from Bethyl Laboratory; PA5-66841 from Thermo) for performing endogenous immunofluorescence experiments, however, none of them were found suitable for immunofluorescence studies. A diffused signal of similar intensity was observed in both control and knock out cells indicating that the antibodies cannot be used for IF studies. So, we were not left with any choice except to perform the co-localization studies with exogenously expressed TRIM16.*

Nevertheless, we performed the western blotting experiments with soluble

New Experiment 4

conditions (Figure 7L). This experiment shows that not only exogenously expressed TRIM16 but also endogenous TRIM16 is enriched in protein aggregates. To further strengthen this finding, we have now performed DSS cross-linking experiments with protein aggregates fraction (insoluble fraction) of cell. The data very clearly show that TRIM16 is present as oligomers (along with oligomers of p62) in the protein aggregates

*/insoluble fraction to show that the **endogenous TRIM16** is enriched in the insoluble protein aggregates fractions along with p62 under stress*

New Experiment 5

fraction (New experiment 5, right panel and Figure 7N) suggesting that endogenous TRIM16 does localize with p62 in protein aggregates.

The endogenous TRIM16 interacts with endogenous LC3B and this interaction is increased on treatment of cells with MG132 (Figure Expanded View Figure 5D). In immunofluorescence experiments, GFP-TRIM16 and LC3B were very nicely co-localized (New experiment 4, above panel) and MG132 increases the number of aggregates and co-localization. Addition of BafilomycinA1 increased the size of aggregates further (New experiment 4 above panel).

Reviewer Question: Is TRIM16 recovered in the detergent-insoluble fraction?

Ans. Yes, the data is presented in the Figure 7L in the manuscript (and also discussed above). In addition, a new DSS cross-linking experiments with protein aggregates fraction (insoluble fraction) is also discussed above (Figure 7N).

Reviewer comment: Authors use TRIM16 reconstitution/overexpression throughout their study but do not comment on whether TRIM16 is itself an aggregation-prone protein.

Ans. We are very sorry for not bringing this point clearly in the manuscript. The reviewer is correct that we didn't comment on the self-aggregation capacity of TRIM16. Figure 7I and 7J and three new experiments (discussed below) further demonstrate the self-aggregation/oligomerization capacity of TRIM16:

1. DSS cross-linking experiments with the protein aggregates fraction shows that TRIM16 forms dimers and higher oligomers and present in insoluble protein aggregates. (New experiment 5, Figure 7N)

2. The strong interaction in pull down experiments with two differently tagged-TRIM16 (Myc and GFP) shows that TRIM16 does have self-oligomerization capacity. (New experiment 6, right panel, Figure 7M)

3. Immunofluorescence experiments with two differently tagged-TRIM16 (Myc and GFP) shows that both tagged protein co-localize completely with each other and form protein aggregates suggesting that TRIM16 has self-oligomerization and self-aggregation capacity. (New experiment 7, right panel below, Expanded View Figure 5E) The result discussion part of manuscript is changed accordingly (page 12, 13 and 15)

2. TRIM16 interacts with p62 and NRF2 via its SPRY domain

- TRIM16 is known to interact with a plethora of proteins (Chauhan et al., 2016, Kimura, Jia et al., 2017). In the current study, the authors show that TRIM16 binds p62, NRF2 and KEAP1. Using co-IP, the authors map the interactions to the C-terminal SPRY

domain of TRIM16 (Fig. 2P-U). Is this interaction direct? It is recommended to use purified TRIM16 (or its SPRY domain) and purified p62 or NRF2 to prove the direct binding. Would TRIM16 compete for NRF2 or p62 binding with KEAP1? It would be important to include a negative control in the direct protein binding assay (e.g., GST alone in the GST pulldown assay) to demonstrate specificity of the TRIM16 interactions.

Ans. Previously, it was shown by Dr. Deretic's and Dr. Terje's group (I am co-author here) that purified GST-p62 does not interact with TRIM16 (Mandell et al., 2014). In very much agreement with this data, we found that under basal conditions, the interaction between endogenous TRIM16 and p62 is very weak (Figure 2N). This interaction is dramatically increased under proteotoxic stress conditions and most likely takes place at protein aggregates (Figure 2N). So, the TRIM16-p62 interaction is not relevant in purified conditions and our data shows that a biological environment and stress conditions (conditions which induce aggregation) are required for this interaction to take place. We have now discussed this in manuscript in result section (page 7 and paragraph 2).

We found that both endogenous and exogenous TRIM16 interacts with NRF2 and p62 (Figure 2N, 2R and 2S). Further to show the specificity of interaction, we performed domain mapping experiment and showed that the SPRY domain of TRIM16 is required for interaction with NRF2 and p62. Furthermore, we showed the importance of TRIM16 SPRY domain in ubiquitination of NRF2 and importantly in protein aggregates formation. Our data in biologically relevant conditions shows the "interaction", the "specificity of interaction" and the "biological importance of the interaction". As the purified conditions are not relevant for studying the TRIM16 and p62/NRF2 biology, all the concerns of reviewers were studied in the biologically relevant conditions.

The questions raised above in point 2 are divided in parts for clarity purpose;

Would TRIM16 compete for NRF2 or p62 binding with KEAP1?

Ans. We have now performed the experiment to determine how TRIM16 affects the interaction between NRF2 and KEAP1 (New experiment 8, Right panel). The data show that TRIM16 reduces KEAP1-NRF2 interaction suggesting that one

of the mechanism by which TRIM16 stabilizes NRF2 is by displacing KEAP1 from NRF2. The experiment is presented in Figure. 3A and 3B and discussed at page 7 paragraph 3.

Does the SPRY domain interact with itself or is that the coiled-coil domain (CCD) only that would support potential TRIM16 oligomerization?

Ans. To answer this, we studied interaction between TRIM16 domains (Myc-tagged) and full length TRIM16 (Flag-tagged). The domain interaction study suggests that CCD is the main player in oligomerization of TRIM16 (New experiment 9, Left panel). The data is presented in Figure 7O.

3. TRIM16 fosters p62 oligomerization. The authors suggest that TRIM16 affects protein aggregate formation in part by fostering p62 oligomerization. Without looking into the possible scaffolding role of TRIM16, they show that TRIM16 reduces the K63-ubiquitylation status of p62 which might lead to increased p62 oligomerization. The authors refer to the study by Pan et al, in which another TRIM family member, TRIM21, promoted K63-linked ubiquitylation within the PB1 domain of p62, which is responsible for p62 oligomerization (Pan, Sun et al., 2016). However, in their own study, the authors do not provide strong evidence for the changes in p62 oligomerization. The Flag/GFP-p62 interaction assay (Fig. 3F), although indicative of the effect, does not provide reliable information. Why would GFP-p62 and Flag-p62 interact in the cell poorly in the first place? The PB1 domain of p62 should ensure intermolecular interaction between p62 molecules attached to different tags. Endogenous NBR1 is the protein that might interfere with the cellular p62 oligomerization assays. NBR1 possesses a PB1 domain that interacts with the PB1 domain in p62 (Kirkin, Lamark et al., 2009). In the least, the potential effect of NBR1 needs to be excluded from the system (e.g., by using p62 mutants that do not bind NBR1 or using NBR1-depleted cells). Biochemical assays using purified proteins might be a resource-intensive alternative to the cellular assays. If TRIM16 binds p62 directly, does this per se promote p62 oligomerization (irrespective of p62 ubiquitylation status)? Results obtained with overexpression of p62 in TRIM16 KO HeLa cells (Fig. 5A-D) do not support this notion so far and are at odds with the authors' preceding finding (Fig. 3F). An experiment using purified TRIM16 and p62 could cast light on this issue.

Ans: We understand that TRIM16-TRIM21 cross talk and p62 oligomerization data is preliminary and does not show thorough understanding of TRIM16-mediated p62 oligomerization. Since understanding the oligomerization of p62 is not the main theme of manuscript (as compared to Pan et al, 2015) and as per the concern of reviewers about "excess data" and "unity of the data", and also after thorough discussion with the editor, the data pertaining to the p62 oligomerization and TRIM21-TRIM16 cross-talk was deemed as being best removed from the revised manuscript. We have noted the reviewer's important points and we will work on them in details in a separate study.

4. TRIM16 activates NRF2 signaling

The authors claim that TRIM16 is found in a complex with p62, KEAP1 and NRF2 (Fig. 2). Of interest, while the TRIM16-KEAP1 interaction was not affected (Fig. S2G), the TRIM16-p62 complex (Fig. 2O) was strengthened and the TRIM16-NRF2 complex was weakened (Fig. 2S vs. Fig. S2D) after the MG132 treatment.

As shown previously, p62 competes with NRF2 for binding to KEAP1, liberating NRF2 from its inhibitory/degradative complex with KEAP1 and

thereby activating the NRF2 signaling (Jain, Lamark et al., 2010, Komatsu, Kurokawa et al., 2010).

So, how do the authors envisage the way TRIM16 would fit into this well-known scheme? Does TRIM16 oust KEAP1 from its complex with NRF2, as is the case for p62? Or, is TRIM16's role merely to bring more p62 into the close proximity with NRF2 (the scaffolding role)?

TRIM16 seems to affect NRF2 ubiquitylation status (Fig. 3A-B). Since the TRIM16-KEAP1 complex is not affected by the MG132 treatment, and KEAP1 is the major E3 ligase for NRF2, it is presently not clear what causes the change in NRF2 ubiquitylation pattern. To claim a TRIM16-mediated activating effect on NRF2, the authors must provide a mechanistic explanation using their own data. Current explanations are speculative and insufficient.

Ans. Our data suggest that TRIM16 activates NRF2 in several ways:

1. TRIM16 reduces NRF2 and KEAP1 interactions (New experiment 8) liberating it from inhibitory complex. Further, TRIM16 increases interaction between KEAP1 and p62, most likely for autophagic sequestration/degradation of KEAP1 (Figure 3D, discussion page 15, paragraph 2).

2. TRIM16 also increases vicinity of NRF2 with p62 (scaffolding role) (New experiment 10, Figure 3C).

3. TRIM16 decreases K48-linked ubiquitination of NRF2 (may be by blocking KEAP1 interaction with NRF2) and increases K63-linked ubiquitination of NRF2. Both events can provide stability to NRF2. The SPRY domain of TRIM16 is important for K63-linked ubiquitination of NRF2, activation of NRF2 regulated genes, ubiquitin conjugation to misfolded proteins and protein aggregates formation suggesting that K63-linked ubiquitination is an important event in NRF2 stability and activation. Taken together, the data suggest that TRIM16 invokes several mechanisms to stabilize and activate NRF2. The data is discussed at page 17 and paragraph 2.

5. TRIM16 drives transcription of NRF2-responsive genes

- The authors show that the transcription of a number of NRF2-responsive genes (upon oxidative/proteotoxic stress) is significantly reduced in TRIM16 KO cells (e.g., Fig. 3L-N). This is suggested to be due to the scaffolding role of TRIM16 in stabilizing NRF2, p62 and protein aggregates. It would therefore be important to test if TRIM16 overexpression alone (in the absence of the proteotoxic or oxidative stress!) is sufficient to induce NRF2 gene response. Presumably, this should not be the case in NRF2 KO cells.

Ans: We overexpressed TRIM16 and performed the qRT-PCR to determine the status of NRF2 regulated genes (New Experiment no. 1). Overexpression of TRIM16 increased the expression of the key NRF2-regulated genes, Ho-1, p62, and Nqo1. This increased expression is blunted in absence of NRF2 (siRNA knock down of NRF2) indicating that this enhanced expression is mediated via NRF2. Thanks to reviewer for asking this important question. These experiment are presented in (Figure 5

H-J) and below.

6. TRIM16 stimulates autophagy

• The authors propose that TRIM16 positively regulates autophagy. This seems to be in line with the previous report (Chauhan et al., 2016). However, it is unclear whether TRIM16 is itself degraded by autophagy. Use of the well-characterized double tag (RFP-GFP) attached to TRIM16 to monitor its lysosomal distribution could help answer this question.

Ans: We cloned TRIM16 as mCherry-YFP double-tagged protein (like RFP-GFP tag). This vector is obtained from Dr. Johansen

Terje's lab (University of Tromso, Norway). The cells overexpressing mCherry-YFP-TRIM16 were subjected to starvation. If TRIM16 puncta's are acidified by lysosomal activity, then the yellow color of TRIM16 puncta's should appear red in starvation conditions due to quenching of YFP fluorescence. We did not observe red puncta's in basal or starvation conditions indicating that TRIM16 is not a target of autophagy (New experiment 11, Expanded View Figure 5H). This finding is further corroborated by western blotting experiments where we did not observe any change in TRIM16 levels when the cells were subjected to starvation in presence or absence of BafilomycinA1 (New experiment 12, Expanded View Figure 5I). Thanks to reviewer for asking this important question.

• If TRIM16 is overexpressed, it does occasionally form aggregates (e.g., Fig. 5K) suggesting its propensity to self-associate. Do these aggregates become larger if autophagy is inhibited (e.g., in bafilomycin A1-treated cells)?

Ans: In presence of BafilomycinA1 alone, the aggregates size does not increase, however, on addition of MG132 alone or MG132 with Bafilomycin A1, the protein aggregates become larger (Figure 7I and 7K) (New experiment 4) and also the number was increased (Figure 7J).

What is the role of the p62/NBR1 system in TRIM16 aggregation? Do TRIM16 aggregates become larger or smaller if p62 and/or NBR1 are absent from the system?

Ans: Interestingly, in p62 knock down cells, the formation of TRIM16 aggregates was significantly reduced (New experiment 13, left panel Expanded View Figure 5F). The data suggest that p62 is required for TRIM16 aggregates formation. Thanks to reviewer for

asking this important question. We can't comment on role of NBR1 in this process as this work is out of scope of current working model of investigation.

• The data on mitophagy are preliminary and can be removed from the manuscript, unless proper dissection of the role of TRIM16 in mitochondrial ubiquitylation or on expression of mitophagy receptors is performed.

Ans: We agree; the data is removed. Thanks for the suggestion.

7. TRIM16 affects cell survival during oxidative and proteotoxic stress

• The use of AS2O3 in cell and xenograft assays, comes a bit as a surprise at the end of the manuscript. All mechanistic studies were performed using H2O2, MG132 and Puromycin. Why not to use the clinically relevant proteosomal inhibitor Velcade/Bortezomib in the cell and xenograft studies to complete the experiments in one line?

Ans. As₂O₃ (like H₂O₂) can induce oxidative stress and

NRF2 anti-oxidant response (Lau et al, 2013; Li et al, 2013; Woo et al, 2002). In terms of clinical relevance, As₂O₃ is currently the most active single drug agent for treatment of acute promyelocytic leukemia (Lengfelder et al, 2012). Further, several studies showed potency of As₂O₃ in treatment of solid tumors (Akhtar et al, 2017; Maeda et al, 2004) and also it is in clinical trial for several other cancers (<https://clinicaltrials.gov>). As₂O₃ doses and its efficacy in tumor xenograft model is very well defined. These all points were taken into consideration before utilizing As₂O₃ in our experiments. In our study, before using it in the xenograft model, it was used (along with H₂O₂, Mg132) in in-vitro cell culture model to understand the role of TRIM16 in protecting cells from stresses. We understand the reviewer concerns and to bring complete study in single line we have determined the role of TRIM16 in As₂O₃ (oxidative stress) induced protein aggregates formation and also in regulation of NRF2, p62 and ubiquitin (New experiment 14 and New experiment 15, Expanded View Figure 11, Figure 5D). The data show that TRIM16 is required for AS₂O₃ induced protein aggregates formation. Further, we found that AS₂O₃ induced NRF2, p62 and Ubiquitin cojugates levels were dramatically less in TRIM16^{KO} cells (New experiment 15, Figure 5D). This data shows that TRIM16 have similar role in stress response whether the agent is As₂O₃ or H₂O₂.

8. Structure of the manuscript

- The Introduction and Discussion sections can be more detailed, while some passages from the Results section need to be moved in the Introduction. Similarly, some speculative passages in the Results section should be moved into the Discussion section.

We have made following changes:

1. *Introduction: The new recently published role of TRIM16 is discussed.*
2. *The speculative parts from results section are removed or moved to discussion.*
3. *The discussion part of manuscript is increased/changed accordingly (page 15, paragraph 2, page 16, paragraph 2)*

Referee #3:

In this study, Jena and co-workers show that TRIM16 plays a dual role in the cellular response to aggregated proteins by stimulating their sequestration in inclusion bodies as well as the clearance of aggregated proteins by macroautophagy. These are new roles for TRIM16 and the authors argue that the p62-NRF2 axis revealed in this study links the anti-oxidative response to ubiquitin-dependent clearance of misfolded proteins. The authors provide an extensive amount of data supporting this model using mostly microscopic and biochemical tools with which they study this process in cell-based systems. Moreover, they also the importance of this response for tumor growth in a xenotransplant mouse model. The finding is novel and the study is detailed and overall convincing.

We are very thankful to the reviewer for praising our work. Now, we have further strengthened this manuscript by utilizing the advice of all of the 3 reviewers. A total of 22 new experiments are performed.

Major concerns

Title and conclusions of the paper. Can the authors exclude the possibility that TRIM16 does not induce degradation of aggresomes by autophagy but the formation of aggresomes by facilitating degradation of protein aggregates or aggregation-prone proteins that would otherwise end up in the aggresome. Note that aggresomes and protein aggregates are not the same thing. If this possibility cannot be concluding the title should be phrased more carefully as it suggests now that the aggresomes themselves are turned over.

Ans: We agree with the reviewer and apologies for the confusion. We have changed the title of the manuscript accordingly. The new title is:

“TRIM16 controls protein aggregates assembly and degradation by modulating p62-NRF2 axis and autophagy”

Fig. 1J Does TRIM16 knockout affect the levels of total ubiquitin conjugates? If so, that could be the reason for a reduction in ubiquitin-positive inclusions in TRIM16 knockout cells. Is there a selective effect on total/soluble/insoluble levels of K48 or K63 chains?

For clarity reason, Reviewer's above set of questions are answered in parts:

Q. Does TRIM16 knockout affect the levels of total ubiquitin conjugates?

Ans: Yes, TRIM16 knockout reduces the levels of total ubiquitin conjugates (Figure 1J, 5D, 5K, 6I and 9H). This was the reason why we thought that TRIM16 might be a regulator of transcription of Ubiquitin pathway genes. We tested and found that TRIM16 indeed regulates transcription of Ubiquitination pathway genes via NRF2 (Figure 5L, Expanded view Figure 3J to 3L, Figure 6B and 6C).

Q. If so, that could be the reason for a reduction in ubiquitin-positive inclusions in TRIM16 knockout cells.

Ans: Yes, the reviewer is correct that the reduction in total ubiquitin conjugation could be one of the major reasons for reduction in ubiquitin-positive aggregates.

Is there a selective effect on total/soluble/insoluble levels of K48 or K63 chains?

Ans: Upon TRIM16 depletion, the K48-linked ubiquitin conjugates are reduced in both soluble and insoluble fractions. However, K63-linked ubiquitin conjugates are reduced only in insoluble fractions (New Experiment 16, Expanded view Figure 3I).

Fig S1M-O. I don't understand the rationale for the choice of ULK1 and Beclin1 as representative substrates for monitoring inclusions. Both proteins are also involved in macroautophagy. Why did the authors select these proteins? It would be better to use an aggregation-prone substrate that is not involved in clearance of misfolded

proteins as a reference.

Ans: The reviewer point is very valid. This data was an additional assay but was not the prime assay on which our conclusions were based and hence was presented in Supplementary Figures in old manuscript version. Since these are confusing results, we have removed this data from the revised manuscript.

Fig 1P. If I am not mistaken, proteostat has an excitation wavelength of around 480 nm. The authors did not mention the Alexa dye that was used in that experiment but I suspect that p62 was stained with an Alexa488 dye (because it is depicted green in the micrograph). If so, there is a serious risk for spectral contamination when combining these dyes because they will be excited with the same wavelength. Careful selection of emission filters may not entirely solve this problem. Therefore, it would be helpful if the authors (for example in the supplementary figures) show micrographs for proteostat and p62 of cells that have only been stained with proteostat or for p62.

Ans: The reviewer is absolutely correct that both Proteostat and Alexa 488 have similar excitation wavelength. However, the emission spectra are very different (Proteostat=610 and Alexa 488 =520). Because of this huge difference, Alexa 488 is always the choice of dye with Proteostat dye for double staining (<http://www.enzolifesciences.com/ENZ-51035/proteostat-aggressive-detection-kit/>). Here are two more references (there are many other papers):

1. Alexa 488 (p62) and Proteostat (Maejima et al, 2013)
2. GFP tagged protein and Proteostat (Kothawala et al, 2012)

Our data also indicate that the p62 fluorescence (Alexa 488) and Proteostat

fluorescence (red) does not overlap (even a bit) in TRIM16^{KO} cells (Figure 1P) suggesting that our condition of capturing the images (emission filters) are not allowing any cross contamination between Alexa 488 and Proteostat.

As suggested by the reviewer, we have now performed the experiment and presented some of the images where cells were either stained with Proteostat or with Alexa 488 (p62) in Expanded View Figure 1O (New Experiment 17).

We are thankful to the reviewer for bringing this point. On careful analysis, we think the reviewer point regarding spectral cross contamination is valid in case of Alexa 633 (Far red- Ub staining in previous manuscript figure 1P) and Proteostat (overlapping emission range). Hence, we have removed Ub micrographs from the Figure 1P and just showing results of Proteostat and p62. This change does not affect any conclusions.

Fig 2M-N. The destabilization of p62 is not very convincing. On how many experiments are the curves shown in Fig 2N based? Include error bars.

Ans: Yes, compared to the effect on NRF2,

the effect on p62 stability is not huge, however, it is consistent. The new graphs are plotted from 3 different experiments (Figure 2M). We have now presented one more repeat experiment in the Expanded Figures 2D and here (Right panel).

Fig 2O. Not only the input but also pull-down of TRIM16 should be shown for the immunoprecipitations. It is not clear for me why the blot is cut after the lanes that show the input for TRIM16.

Ans: The full blot of pull down in the same experiment re-probed with TRIM16 is shown in left panel. In previous version of figure, we just showed the input which was run separately as the inputs were not very visible in this blot. We

have now added this blot also in the main figure (Figure 2N).

Fig 3A, B, D, F. There are several controls missing in these pulldown experiments.

First, these experiments should include pulldowns from lysates of cells that express plain GFP (for GFP-NRF2) or mock transfected cells (for epitope tagged p62) to determine the specificity of the pulldown. Second, the western blot should not only have been shown for the input of the precipitated protein (GFP-NRF2 or epitope tagged p62) but also for the pulldown. Third, molecular weight markers should be depicted. This is of particular interest for Fig 3A and B as it would allow evaluating if the ubiquitin smear localizes at the correct molecular weight for ubiquitylated GFP-NRF2.

Ans: Fig 3A and 3B (Now Figure 4A and 4B): The control experiments were performed where we replaced GFP-NRF2 with GFP to determine the specificity of the pull downs. The Data show that the effect of TRIM16 on NRF2 ubiquitination is specific (New Experiment 18, 19, Appendix Figure S1A and S1B). GFP-NRF2 in pulldown is now shown in Figure 4A and 4B.

Figure 3D and 3F. These experiments are removed in revised manuscript:

These experiment were related to p62 oligomerization. Since this part of the manuscript is weak and does not give thorough understanding of this phenomenon and also as per the concern of reviewer 1 about streamlining of the data, after discussion with the editor, the data pertaining to the to the p62 oligomerization and TRIM21-TRIM16 cross-talk is removed from revised version of manuscript.

The molecular weight markers are now depicted in every blot of the Figures.

Fig 3D. p62 is an ubiquitin-binding protein and there is a risk that the signal on the ubiquitin blot comes from ubiquitylated proteins that were co-immunoprecipitated with FLAG-p62 and not from ubiquitylated p62. Was the sample denatured before the immunoprecipitation was performed? Can the authors exclude that the signal comes from ubiquitylated proteins bound to p62?
Ans: We agree with the reviewer that we can't exclude the possibility that the signal is entirely due to the ubiquitination of p62. To rectify this problem, we have now performed reverse IP where we have performed pull down with HA (ubiquitin) antibody and Western blotting is performed with flag antibody (p62). This experiment tells us specifically the status of ubiquitinated-p62 but not of the ubiquitination of its binding partners. The data clearly indicates that

K63-linked ubiquitination of p62 is dramatically reduced in presence of TRIM16 (Experiment 20). However, for the reason described above, Figure 3D is removed from the revised manuscript and thus this new experiment is performed for the review purpose only.

Fig 3Q. It appears that p62 immunostaining signal is also dramatically decreased when *UBB* or *Ube2N* are depleted. Is there a reduction in the total levels of p62 when these genes are knocked down? This can be checked by western blotting.

Ans. Yes, reviewer is absolutely correct, upon depleting of Ubb and Ube2n the total amount of p62 was decreased in western blots (Experiment 21, Figure 5O).

Fig 4I. I guess that the difference in the molecular weight of endogenous NRF2 and myc-NRF2 is too little to detect this as independent bands in the Nrf2 blot.

Even if this is the case, I would still expect an increase in the NRF2 band intensity because of the overexpression of myc-NRF2. Please explain why this is not observed. It would also be helpful to show a larger part of the NRF2 blot so that the GFP-NRF2 can be seen in the same NRF2-probed blot. Also here it would be helpful if molecular weight markers are included.

Ans: We have observed this now several times that NRF2 antibodies (Abcam#62352 or CST#12721s) stains very weakly with

overexpressed tagged versions of NRF2 or sometimes not at all. Due to this reason, it's a very normal practice in the lab to re-probe the same blot with Myc and/or with GFP (as per requirement of experiment) to get clear bands of tagged-NRF2. We have now repeated the experiment under similar conditions and tried to get the Myc-NRF2 and GFP-NRF2 band with NRF2 antibodies in the same blots (**Experiment 22**). Although, blot is not that great but we can observe both tagged and un-tagged versions together. In terms of results, this experiment is perfect repeat of previous experiments including the one which is presented in the manuscript. The new experiment is presented here for review purpose only.

Fig 5N. The patterns for ULK1 and ATG16L1 are very different. Isn't that surprising since they are both acting on early autophagosomes and are therefore expected to co-localize?

The reviewer is correct. The pattern of co-localization is different. These experiments were performed in presence of proteasomal inhibitor MG132 so as to induce protein aggregates formation. In case of ULK1, most of the MG132 induced aggregates are positive for ULK1, TRIM16, and Ubiquitin. However, not all the structure stained by ATG16L/Ub are TRIM16 positive. The possibilities are that these ATG16L structure which co-localize with Ub but not with TRIM16 are not protein aggregates. This is difficult for us to predict that what these structure are if they are not protein aggregates. This need further investigation and might be interesting to look into, however, it is not within the scope of the current study.

TRIM16 associates with proteins like p62 that localize in the autophagosome and are degraded in the process of targeting substrates for degradation. Is TRIM16 degraded by autophagy?

Ans: As suggested by reviewer 2, we cloned TRIM16 with tandem-tag mCherry-YFP to find whether TRIM16 is a target of autophagy. The cells

overexpressing mCherry-YFP-TRIM16 were subjected to starvation. If

TRIM16 puncta's are acidified by lysosomal activity then the yellow color of TRIM16 puncta's should

appear red under starvation conditions due to quenching of YFP fluorescence. We did not observe any red puncta's in basal or starvation conditions indicating that TRIM16 is not a target of autophagy (**New experiment 11**, Expanded View Figure 5H). This finding is further corroborated by western blotting experiments where we did not observe any change in TRIM16 levels when the cells were subjected to starvation in

presence or absence of BafilomycinA1 (New experiment 12, Expanded View Figure 5I). Thanks to the reviewer for asking this important question.

Minor concerns

Abstract. The opening sentence is too generic. The authors talk about "a protective effect of assembly of aggregates from misfolded proteins". While it is rather well established that the sequestration of protein aggregates in inclusion bodies/aggregates has a beneficial effect, the actual aggregation of misfolded proteins may well be linked to their toxicity and not protective. *The lines are changed accordingly. Thanks for the suggestion.*

"TRIM16 interacts, stabilizes, affects K63-linked ubiquitination, and increases the oligomerization capacity of p62". Grammar incorrect, sentence unclear.

The line is removed.

Introduction

"Normally, these misfolded proteins are ubiquitinated and are degraded by the proteasome system or chaperone-mediated autophagy (Kraft et al., 2010)." What about macroautophagy? The review that is cited discusses primarily clearance of misfolded proteins by ubiquitin-selective (macro)autophagy. It is better to cite the original papers but if necessary and reviews need to be cited they have to be correct. Here references/reviews on the role of CMA and the proteasome in clearance of misfolded should be included.

We apologize. The changes are made accordingly. Thanks for the suggestions.

A few recent papers on TRIM16 and (secretory) autophagy that appear to be relevant

for this study are not cited/discussed. In particular, Fraiberg and Elazar, 2016,

Developmental Cell; Kimura et al, Autophagy, 2017; Kumar et al, Autophagy, 2017 and

Kimura et al, EMBO Journal, 2017.

We apologize. The papers are cited now in introduction and results at Page 5, paragraph 1. Thanks for the suggestion.

Results

"dot's" should be dots.

The changes are made accordingly in whole manuscript.

Discussion

"autophagosomes" should be autophagosomes.

We apologize. The changes are made accordingly. Thanks.

"The poly-ubiquitin proteins in human are encoded by Ubb and Ubc genes". I would rather talk about ubiquitin precursor proteins so that it is not confused polyubiquitin conjugates. Note that there are two more ubiquitin

precursor proteins in the human genome.

We agree, the changes are made accordingly. Page 15, Paragraph 3.

Please check for grammar and spelling.

We have checked and corrected for spellings and grammar. Thanks.

References

Akhtar A, Xiaoyan Wang S, Ghali L, Bell C, Wen X (2017) Recent advances in arsenic trioxide encapsulated nanoparticles as drug delivery agents to solid cancers. *Journal of biomedical research* **31**: 177-188

Kothawala A, Kilpatrick K, Novoa JA, Segatori L (2012) Quantitative analysis of alpha-synuclein solubility in living cells using split GFP complementation. *PloS one* **7**: e43505

Lau A, Zheng Y, Tao S, Wang H, Whitman SA, White E, Zhang DD (2013) Arsenic inhibits autophagic flux, activating the Nrf2-Keap1 pathway in a p62-dependent manner. *Molecular and cellular biology* **33**: 2436-2446

Lengfelder E, Hofmann WK, Nowak D (2012) Impact of arsenic trioxide in the treatment of acute promyelocytic leukemia. *Leukemia* **26**: 433-442

Li B, Li X, Zhu B, Zhang X, Wang Y, Xu Y, Wang H, Hou Y, Zheng Q, Sun G (2013) Sodium arsenite induced reactive oxygen species generation, nuclear factor (erythroid-2 related) factor 2 activation, heme oxygenase-1 expression, and glutathione elevation in Chang human hepatocytes. *Environmental toxicology* **28**: 401-410

Maeda H, Hori S, Ohizumi H, Segawa T, Kakehi Y, Ogawa O, Kakizuka A (2004) Effective treatment of advanced solid tumors by the combination of arsenic trioxide and L-buthionine-sulfoximine. *Cell Death Differ* **11**: 737-746

Maejima Y, Kyoji S, Zhai P, Liu T, Li H, Ivessa A, Sciarretta S, Del Re DP, Zablocki DK, Hsu CP, Lim DS, Isobe M, Sadoshima J (2013) Mst1 inhibits autophagy by promoting the interaction between Beclin1 and Bcl-2. *Nature medicine* **19**: 1478-1488

Woo SH, Park IC, Park MJ, Lee HC, Lee SJ, Chun YJ, Lee SH, Hong SI, Rhee CH (2002) Arsenic trioxide induces apoptosis through a reactive oxygen species-dependent pathway and loss of mitochondrial membrane potential in HeLa cells. *International journal of oncology* **21**: 57-63

Thank you for submitting your revised manuscript for our consideration. We have now received the re-reviews from all three original referees, and I am pleased to inform you that they all consider the manuscript significantly improved and the majority of originally raised issues addressed. Nevertheless, referee 3 retains one major concern regarding the conclusion that NRF2 is directly ubiquitinated by TRIM16. Since this well-taken issue appears relevant for the overall interpretation of the study, I feel it will be important to further address this remaining point through further experiments along the lines suggested by the reviewer. I am therefore returning the manuscript to you once more for an exceptional round of additional revision, during which you should also take care of the important presentational points mentioned by referee 2.

REFEREE REPORTS

Referee #1:

I thank the authors for their effort. The authors have adequately responded to my questions. I think that the manuscript is now almost ready for publication.

Referee #2:

Generally, I am satisfied with the revision made by the authors. They addressed the majority of my concerns and provided a lot of additional data. I am a bit concerned with the lack of clarity on how the TRIM16 SPRY domain interacts with p62/SQSTM1 if these two proteins do not interact directly in vitro (Mandell et 2014). However, the cell-based data, which are plentiful, suggest that there should be an additional mechanism involved (such as post-translational modification of the interaction partners). Therefore, I favor publication of this revised paper.

The paper still needs proof-reading by a native speaker and that some parts of the results are moved into Introduction and/or Discussion. The Results sections often contain long passages on the background information whose place is in the introduction.

Referee #3:

The authors have addressed most of my concerns but one critical point remains. This relates to the claim that TRIM16 regulates ubiquitylation of NRF2, which is shown in Figure 4. I have serious doubts whether the authors have conclusively shown that NRF2 ubiquitylation is stimulated by TRIM16 and whether this is due to ubiquitin ligase activity of TRIM16.

There are three main reasons for this concern. First, now that the molecular weight markers are included in the Western blots, it is clear that the ubiquitin conjugates show a smear that is rather homogenous in intensity from <75 kD to >250 kD (Fig 4A-C). This is hard to reconcile with this being ubiquitylated GFP-tagged NRF2 as this runs at a molecular weight of just below 150 kD. To me this suggests that the ubiquitylated proteins that are detected in the GFP-NRF2 pulldown are not ubiquitylated GFP-NRF2. Second, the levels of what is supposed to be ubiquitylated GFP-NRF2 are also stimulated by a fragment consisting of only the SPRY domain of TRIM16. The authors argue that this is consistent with an earlier study by Bell and co-workers that showed that the SPRY domain plays a role in its ubiquitylation activity. However, in this study is also shown that the B boxes, which are missing in this fragment, are required for the ubiquitin ligase activity of TRIM16. Hence, according to the earlier study the SPRY domain should lack ubiquitin ligase activity. Third, I assume that the pulldown of GFP-NRF2 is performed under native conditions and therefore it cannot be excluded that the ubiquitin conjugates that are detected in the pulldown are not ubiquitin molecules conjugated to NRF2 but rather ubiquitylated proteins that are directly or indirectly bound to NRF2.

In my opinion, these data are more consistent with an effect of TRIM16 in the levels of ubiquitin conjugates that are bound to NRF2. This is maybe not unexpected since the authors show that TRIM16, NRF2 and p62 form a complex and p62 binds strongly ubiquitylated proteins. Thus, these experiments may instead be consistent with the SPRY domain being important for the interaction between NRF2 and p62 (or another ubiquitin binding protein) as the authors show in Figure 2.

To support the claim that TRIM16 stimulates ubiquitylation of GFP-NRF2 the authors should show that the ubiquitin conjugates are indeed ubiquitylated NRF2 and not other proteins. This can for example be done by co-expression of His-tagged ubiquitin and GFP-NRF2, pull down the His-tagged ubiquitin under denaturing conditions followed by detection of GFP-NRF2. If these experiments would support my alternative interpretation, the manuscript and model should be amended accordingly by removing the claim that TRIM16 stimulates ubiquitylation of NRF2.

Re: EMBOJ-2017-98358R

TRIM16 controls protein aggregate assembly and degradation by modulating p62-NRF2 axis and autophagy

Referee #1:

I thank the authors for their effort. The authors have adequately responded to my questions. I think that the manuscript is now almost ready for publication.

Referee #2:

Generally, I am satisfied with the revision made by the authors. They addressed the majority of my concerns and provided a lot of additional data. I am a bit concerned with the lack of clarity on how the TRIM16 SPRY domain interacts with p62/SQSTM1 if these two proteins do not interact directly in vitro (Mandell et 2014). However, the cell-based data, which are plentiful, suggest that there should be an additional mechanism involved (such as post-translational modification of the interaction partners). Therefore, I favor publication of this revised paper.

The paper still needs proof-reading by a native speaker and that some parts of the results are moved into Introduction and/or Discussion. The Results sections often contain long passages on the background information whose place is in the introduction.

Thanks for the suggestions. The paper is now proof-read by a native speaker. The long passages in the results section are shortened or moved to Introduction section:

- 1. Page 5, paragraph 3, the introductory lines are shortened.*
- 2. Page 6, paragraph 3, the introductory passage is shortened, and background information is moved to 3rd paragraph of introduction.*
- 3. Page 10, paragraph 3, the introductory passage is shortened, and background information is moved to the 4th paragraph of introduction.*

Referee #3:

The authors have addressed most of my concerns, but one critical point remains. This relates to the claim that TRIM16 regulates ubiquitylation of NRF2, which is shown in Figure 4. I have serious doubts whether the authors have conclusively shown that NRF2 ubiquitylation is stimulated by TRIM16 and whether this is due to ubiquitin ligase activity of TRIM16.

There are three main reasons for this concern. First, now that the molecular weight markers are included in the Western blots, it is clear that the ubiquitin conjugates show a smear that is rather homogenous in intensity from <75 kD to >250 kD (Fig 4A-C). This is hard to reconcile with this being ubiquitylated GFP-tagged NRF2 as this runs at a molecular weight of just below 150 kD. To me this suggests that the ubiquitylated proteins that are detected in the GFP-NRF2 pulldown are not ubiquitylated GFP-NRF2. Second, the levels of what is supposed to be ubiquitylated GFP-NRF2 are also stimulated by a fragment consisting of only the SPRY

domain of TRIM16. The authors argue that this is consistent with an earlier study by Bell and co-workers that showed that the SPRY domain plays a role in its ubiquitylation activity. However, in this study is also shown that the B boxes, which are missing in this fragment, are required for the ubiquitin ligase activity of TRIM16. Hence, according to the earlier study the SPRY domain should lack ubiquitin ligase activity. Third, I assume that the pull-down of GFP-NRF2 is performed under native conditions and therefore it cannot be excluded that the ubiquitin conjugates that are detected in the pull-down are not ubiquitin molecules conjugated to NRF2 but rather ubiquitylated proteins that are directly or indirectly bound to NRF2.

In my opinion, these data are more consistent with an effect of TRIM16 in the levels of ubiquitin conjugates that are bound to NRF2. This is maybe not unexpected since the authors show that TRIM16, NRF2 and p62 form a complex and p62 binds strongly ubiquitylated proteins. Thus, these experiments may instead be consistent with the SPRY domain being important for the interaction between NRF2 and p62 (or another ubiquitin binding protein) as the authors show in Figure 2.

To support the claim that TRIM16 stimulates ubiquitylation of GFP-NRF2 the authors should show that the ubiquitin conjugates are indeed ubiquitylated NRF2 and not other proteins. This can for example be done by co-expression of His-tagged ubiquitin and GFP-NRF2, pull down the His-tagged ubiquitin under denaturing conditions followed by detection of GFP-NRF2. If these experiments would support my alternative interpretation, the manuscript and model should be amended accordingly by removing the claim that TRIM16 stimulates ubiquitylation of NRF2.

We are very thankful to the reviewer for raising this point and asking us to remove the grey area. The reasoning given by the reviewer is very convincing, so we started the experiments with the similar vision and thoughts as of the reviewer's. We performed a number of experiments to address this point very clearly.

1. We cloned His-tagged Ub-K63 (all lysine mutated except at 63 position). The cells were transfected with His-UB-K63 along with GFP-NRF2 in absence and presence of TRIM16. The lysis and also the pull-down of His-Ub-K63 with Ni-NTA beads were performed in denaturing conditions (6M Guanidine hydrochloride containing lysis buffer). The westerns blots were developed with GFP (for ubiquitinated GFP-NRF2). The result clearly shows that the K63-linked ubiquitination of NRF2 is increased in the presence of TRIM16. This data shows that TRIM16 enhances Ubiquitin conjugation with NRF2. These results are discussed and presented on page 8 paragraph 2, Figure 4D and also panel A here.

2. We performed the same experiment as above in non-denaturing conditions also. The results again show that the amount of K63-linked ubiquitinated NRF2 is increased in presence of TRIM16. Since, in these experiments, the affinity pull-down assay is performed for His-tagged

ubiquitin, the smear in western blot represent ubiquitinated NRF2 only. These results are presented at page 8 paragraph 2, Figure 4E and also panel B here.

3. We would like to point out that the smear below 150 KDa could be due to either ubiquitinated isoforms (low molecular weight isoforms) of NRF2 or ubiquitinated degraded NRF2. The degradation of NRF2 is often seen when NRF2 is overexpressed.

4. Keeping point 3 in mind, we repeated the original K63-linked ubiquitination assays (Figure 4B) very cautiously so that the degradation of NRF2 is minimal. For this, we reduced the incubation time after the transfection, increased the amount of protease inhibitor in lysis buffer and loaded the immunoprecipitate's into the SDS-Gel without storing them in -80. This time we performed reverse IP where we pull-down with HA antibody (HA-UB) and probed with GFP antibody (GFP-NRF2). The data clearly shows that TRIM16 increases K63-linked ubiquitination of NRF2 and the ubiquitination smear is observed above 150 kDa (nothing below). This result is presented at page 8 paragraph 2, Figure 4C and also panel C here.

increases K63-linked ubiquitination of NRF2 and the ubiquitination smear is observed above 150 kDa (nothing below). This result is presented at page 8 paragraph 2, Figure 4C and also panel C here.

5. Lastly, to be clear on the experiment about the TRIM16 domain/s important for ubiquitination of NRF2, we repeated the whole experiment again in condition mentioned in point 3. The smear of NRF2 ubiquitination is mostly observed above 150 kDa suggesting that the smear (75-150 kDa) which we observed in our previous experiments (Figure 4A-4C in the previous version) could be due to the ubiquitinated degraded products of NRF2. We have replaced the Figure 4C (in previous version of manuscript) with this new experiment (now presented in Figure 4E). Although the low molecular weight smear (75-150 KDa) is greatly reduced in new experiment, the results remain the same that both SPRY and B-Box domain are important for TRIM16-mediated ubiquitination of NRF2. These results are presented at page 9 paragraph 3, Figure 4E and also panel A in Right. Taken together, we are now sure that TRIM16 increases the K63-linked ubiquitination of

NRF2. We are very thankful to the reviewer for asking this question and resolving the issue.

Thank you for submitting your re-revised manuscript for our consideration. I have now had a chance to carefully look through your responses and additional data, and I appreciate that they adequately address referee 3's remaining concern and now support that TRIM16 may directly ubiquitinate NRF2. In this light, I am happy to say that there are no more scientific objections towards publication in The EMBO Journal.

Corresponding Author Name: Santosh Chauhan

Journal Submitted to: EMBO J

Manuscript Number: EMBOJ-2017-98358R